



# Uncertainty assessment for 3D geologic modeling of fault zones based on geologic inputs and prior knowledge

Ashton Krajnovich[1], Wendy Zhou[1], and Marte Gutierrez[2]

[1]Department of Geology and Geological Engineering, Colorado School of Mines, 1516 Illinois St, Golden, CO, 80401, USA
[2]Department of Civil and Environmental Engineering, Colorado School of Mines, 1012 14th St, Golden, CO, 80401, USA

**Correspondence:** Ashton Krajnovich (akrajnov@mines.edu)

**Abstract.** Characterizing the zone of damaged and altered rock surrounding a fault surface is highly relevant to geotechnical and reservoir engineering works in the subsurface. Evaluating the uncertainty associated with 3D geologic modeling of these fault zones is made possible using the popular and flexible input-based uncertainty propagation approach to geologic model uncertainty assessment – termed Monte Carlo simulation for uncertainty propagation (MCUP). To satisfy the automation re-

quirements of MCUP while still preserving the key geometry of fault zones in the subsurface, a clear and straightforward modeling approach is developed based on four geologic inputs used in implicit geologic modeling algorithms (surface trace, structural orientation, vertical termination depth and fault zone thickness). The rationale applied to identifying and characterizing the various sources of uncertainty affecting each input are explored and provided using open-source codes. In considering these sources of uncertainty, a novel model formulation is implemented using prior geologic knowledge (i.e., empirical and

theoretical relationships) to parameterize modeling inputs which are typically subjectively interpreted by the modeler (e.g., vertical termination depth of fault zones). Additionally, the application of anisotropic spherical distributions to modeling disparate levels of information available regarding a fault zone's dip azimuth and dip angle is demonstrated, providing improved control over the structural orientation uncertainty envelope. The MCUP formulation developed is applied to a simple geologic model built from historically available geologic mapping data to assess the independent sensitivity of each modeling input

on the combined model uncertainty, revealing that vertical termination depth and structural orientation uncertainty dominate model uncertainty at depth while surface trace uncertainty dominates model uncertainty near the ground surface. The method is also successfully applied to a more complex model containing intersecting major and minor fault zones. The impacts of the model parameterization choices, the fault zone modeling approach and the effects of fault zone interactions on the final geologic model uncertainty assessment are discussed.

## 1 Introduction

Three-dimensional (3D) geologic models are becoming the state of the art for the prediction and communication of subsurface geology in a wide range of projects (Turner and Gable, 2007) including regional geologic characterization (Stafleu et al., 2012; Waters et al., 2015); natural resource exploration (Zhou et al., 2007; Zhou, 2009; Zhou et al., 2015); structural geology (Bond et al., 2015; Ailleres et al., 2019); geotechnical site characterization (Thum and De Paoli, 2015; Zhu et al., 2013); geophysics





(Guillen et al., 2008; Høyer et al., 2015); hydrology (Watson et al., 2015); and mining (Wellmann et al., 2018; Yang et al., 2019). The recent widespread adoption of flexible, implicit 3D geologic modeling algorithms (Cowan et al., 2003; Calcagno et al., 2008; Guillen et al., 2008; Jessell et al., 2014; Hillier et al., 2014, 2017) is leading the field of 3D geologic modeling away from the creation of static models based on a single, best interpretation and towards stochastic geologic modeling with quantified uncertainty (Caumon, 2010). Understanding the uncertainty of a 3D geologic model provides not only a measure

of model quality to an end user (Turner and Gable, 2007; Walker et al., 2003; Stamm et al., 2019), but also aids the geologist during model creation by analyzing the quality of input data and highlighting the impacts of subjective prior knowledge and interpretations (Bond, 2015; Wood and Curtis, 2004; Jessell et al., 2018). As the use of 3D geologic modeling continues to grow, novel methods for assessing the uncertainty of various aspects of geologic models is pertinent.

Because a single model conveys no information regarding its uncertainties (Wellmann and Caumon, 2018), multiple realiza-

tion approaches are becoming a popular method for assessing geologic model uncertainty (Wellmann et al., 2010; Wellmann and Regenauer-Lieb, 2012; Pakyuz-Charrier et al., 2018a, b; Pakyuz-charrier et al., 2019; Jessell et al., 2014, 2018; Lindsay et al., 2013; de la Varga and Wellmann, 2016; de la Varga et al., 2019; Schweizer et al., 2017; Thiele et al., 2016; Yang et al., 2019; Schneeberger et al., 2017). Recently, a Monte-Carlo simulation, input uncertainty propagation method, commonly known as Monte Carlo simulation for uncertainty propagation (MCUP), has grown into a widely used method for 3D geologic

model uncertainty assessment (Wellmann and Caumon, 2018). The method focuses on the impact of uncertainty in geologic modeling inputs on a 3D geologic model by generating a set of model realizations based on perturbations in selected modeling inputs, sampled using Monte Carlo simulation algorithms. MCUP is flexible, allowing for a wide variety of uncertainty sources affecting various geologic modeling inputs to be quantified by the user and propagated into the 3D geologic model.

While growing in popularity, the field of geologic model uncertainty assessment remains a developing one and the application

of MCUP to new, practical problems requires unique model formulations. The development of novel MCUP formulations to address specific aspects of 3D geologic modeling will lead to growth in the field not only by broadening the usability of the method, but also by advancing the understanding of the method's strengths and limitations. In addition to assessing the uncertainty of a single geologic model, MCUP naturally fits into Bayesian inference schemes (de la Varga and Wellmann, 2016; Salvatier et al., 2016; Scalzo et al., 2019; Thiele et al., 2019), allowing for future refinement of model uncertainty as new

information is made available.

## 2   Model implementation

This study expands the use of MCUP to a new aspect of geologic modeling – fault zones, or the localized volume of fractured and displaced rock surrounding a finite fault surface, typically composed of a fault core and a damage zone (Caine et al., 1996; Childs et al., 2009; Peacock et al., 2016; Choi et al., 2016). Fault zones introduce regions of altered geotechnical strength and

hydraulic permeability into the surrounding in-tact rockmass and are therefore of major importance to geological engineering projects that rely on accurate assessments of subsurface rock properties (e.g., tunnels, mines). While faults have been the focus of a significant amount of recent geologic modeling research (Røe et al., 2014; Cherpeau et al., 2010; Cherpeau and Caumon,





2015; Aydin and Caers, 2017), these works have focused on modeling fault surfaces directly rather than modeling the 3D geometry of fault zones. Detailed modeling of the 3D geometry of fault zones can improve the understanding of faults' impacts

on geotechnical and reservoir engineering projects due to the fact that variations in fault zone thickness or composition can greatly alter the mechanical and hydrological behavior of a fault, e.g., its sealing potential (Caine et al., 1996; Fredman et al., 2008; Manzocchi et al., 2010). By an in-depth search of the literature, as of yet there is no dedicated approach to characterizing the uncertainty of fault zones in 3D geologic models.

Fault zones may be irregular in shape, creating complex geometries which are difficult to characterize quantitatively (Torabi

et al., 2019a, b). Peacock et al. (2016) provide a detailed list of the various types of damage zones and intersecting fault networks that comprise the general term "fault zone". The inherent complexity of fault zone structure makes their precise modeling intractable in an automated MCUP formulation. A simplified approach to modeling fault zones in 3D geologic models is developed in this study based on the key elements defining fault zone geometry at a practical level of detail.

The proposed workflow for modeling fault zones is provided in Figure 1. The workflow combines observations from a

geologic map with prior knowledge from the literature to approximate the 3D geometry of subsurface fault zones. The implicit geologic modeling software Leapfrog Works, a software specially designed to support creation of subsurface geological models, is used in this study. The 3D fault zone is modeled in Leapfrog Works from four inputs – surface trace (polyline), structural orientation (dip/dip azimuth vector), fault zone thickness (scalar distance function) and vertical termination depth (discretized surfaces). The modeling approach preserves the essential 3D geometry of fault zones in the subsurface while

providing sufficient generalization to fit into an MCUP formulation for uncertainty assessment.

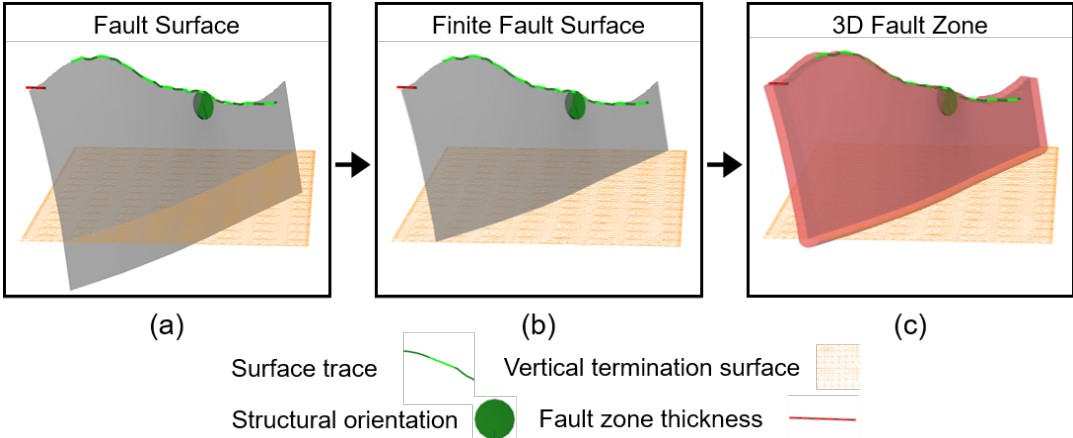

**Figure 1.** The proposed fault zone modeling workflow implemented includes (a) modeling the central fault surface from a polyline and structural orientation, (b) terminating the fault surface on a predefined vertical termination surface and (c) defining the 3D fault zone volume using a distance function from the central fault surface (Krajnovich et al., 2020a).

The popular implicit geologic modeling method was chosen for modeling the uncertainty of fault zones due to its ability to directly incorporate structural orientation data to modeling geologic structures. Leapfrog Works uses radial basis functions



(RBFs) to efficiently interpolate the scalar fields describing implicit geologic surfaces (Seequent, 2014). Use of RBFs in implicit geologic modeling has developed in recent years to rival co-kriging based methods (Hillier et al., 2014, 2017).

## 2.1 MCUP formulation

The MCUP formulation begins with the careful selection of key geologic modeling inputs for perturbation (Figure 1). The set of geologic modeling inputs are characterized using probability distributions chosen and parameterized based on the believed and/or observed uncertainties in the input variables. Monte Carlo simulations independently explore the uncertainty space of each input, generating a set of input realizations which are propagated into 3D geologic models through the use of an automated implicit geologic modeling algorithm. From the set of geologic model realizations, the commonly used Shannon information entropy metric (Shannon, 1948) allows for quantifying the uncertainty about modeled geologic structures (Wellmann and Regenauer-Lieb, 2012). This overview is provided merely to introduce the reader to the general idea of MCUP as applied to implicit geologic modeling; the reader is referred to the book by Wellmann and Caumon (2018) for a more thorough review of the MCUP and implicit modeling methods' conceptual bases.

The same flexibility that allows MCUP to be effectively formulated for nearly any geologic modeling problem is also a potential susceptibility of MCUP – that the model formulation and input uncertainties must be predefined by the user. This can lead to potential over- or under-estimation and biases in the uncertainty assessment performed with MCUP due to inappropriate selection of modeling inputs or incorrect parameterization of input probability distributions (de la Varga and Wellmann, 2016; Wellmann and Caumon, 2018; Pakyuz-Charrier et al., 2018b). Following the school of thought reviewed by Nearing et al. (2016), the modeler must ask questions along the lines of:

- What inputs control the geometry of the modeled structure?

- How can these inputs be defined probabilistically?

- What information is available to characterize each source of input uncertainty?

For hard data inputs (e.g., observed contacts, structural orientation measurements), the answers to these questions are relatively well-established using measures of variance and deviation to directly characterize uncertainties (Caers, 2011; Wellmann and Caumon, 2018). On the other hand, for subjective inputs to geologic models (e.g., interpreted fault terminations), there are generally two approaches with which the subjective uncertainties can be characterized. The first is describing and quantifying the uncertainty associated with prior knowledge by utilizing believed theoretical or empirical relationships to quantify the reasoning behind a subjective geologic interpretation (Wood and Curtis, 2004). The second method, operating within a Bayesian inference scheme, is to incorporate the prior knowledge as a likelihood function to validate – or rather, as Tarantola (2006) states, invalidate – model realizations. This study focuses on the first method of applying prior knowledge from published structural geology literature (Torabi et al., 2019a) to parameterize the reasoning behind subjective inputs used for modeling fault zones in the MCUP formulation. This approach is demonstrated effectively in this study when considering the vertical termination depth of fault surfaces (Section 4.3).





Having identified the geologic modeling inputs controlling the geometry of the modeled fault zone and appropriate meth-
ods for defining these inputs probabilistically, Figure 2 shows the careful consideration of the various sources of uncertainty
affecting each input.

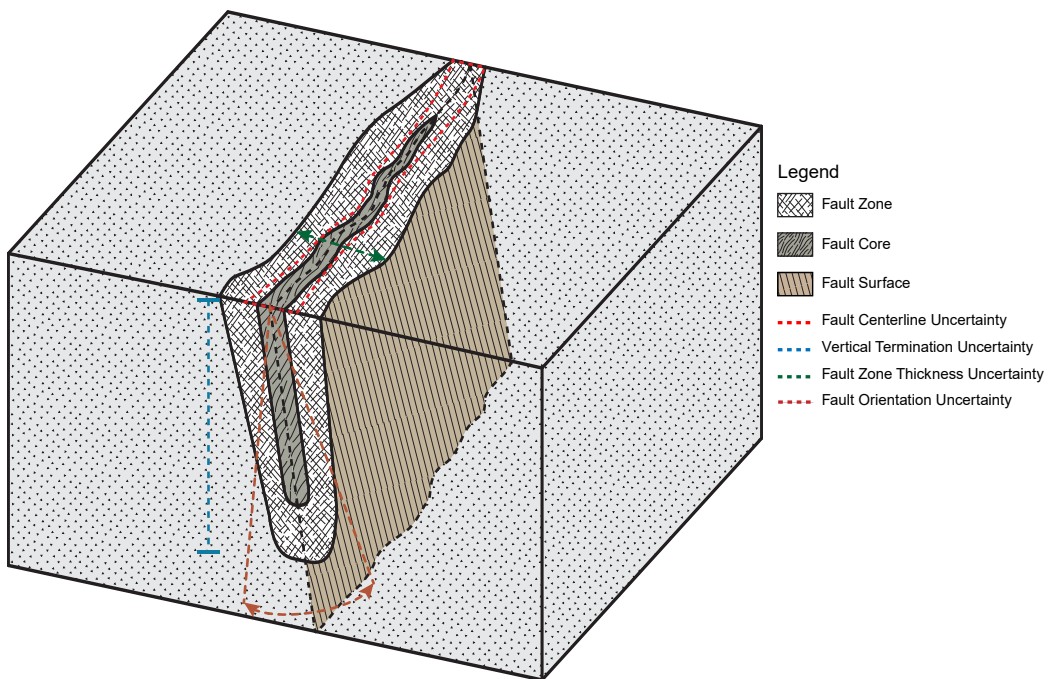

**Figure 2.** A schematic showing the sources of uncertainty influencing the 3D geometry of a fault zone in the subsurface. Modified from
Krajnovich et al. (2020a).

In the 3D geologic model, the geometry and extent of fault zones in the subsurface often are approximated from limited
information available. While useful for conceptualizing subsurface geology and guiding future investigations, the lack of
detailed data contributes significant sources of uncertainty to the 3D geologic model (Figure 2). This study focuses on and
provides guidelines for performing realistic uncertainty assessments when creating 3D geologic models of fault zones from
limited, preliminary investigation data (e.g., a geologic map), further demonstrating how prior knowledge is used to characterize
uncertainty about inputs which are typically subjective (i.e., vertical termination depth). While formulated for a case with
limited data, the developed MCUP formulation allows for accommodating additional observations (e.g., outcrop study) without
significant reparameterization.

Moving forward, Section 3 details the appropriate selection and parameterization of probability distributions to accurately
characterize input uncertainties for various the data types used in 3D geologic modeling of fault zones. Section 4 highlights
the considerations for characterizing each modeling input's uncertainty, and references the Python code written for performing
and assessing the input perturbations on a single fault zone model built from a geologic map in the Rocky Mountains of





Colorado, USA. Section 5 demonstrates the MCUP formulation applied to a more complex, fault network model to assess how the method scales and investigate the interaction of intersecting fault zones. Section 6 discusses the observed results of each model, highlighting key contributions of the MCUP formulation including the impact of different uncertainty parameterizations and guidelines for future model refinement. Section 7 reviews the MCUP implementation for fault zone modeling and reiterates the importance of the rationale used, concluding with recommendations for future work.

## 3 Probability distributions for MCUP


The MCUP formulation developed requires additional advancements in the selection and parameterization of probability distributions used for characterizing the uncertainty of objective and subjective geologic modeling inputs. The selection of an appropriate probability distribution type from the various distributions available for modeling involves the consideration of two factors: the data type and the level of knowledge about the input. Geologic data may be discrete (e.g., lithological cat-

egories) or continuous (e.g., thickness), with various probability distributions available to characterize both data types (e.g., Normal, Uniform, Log-normal, Binomial – see Gelman et al. (2004)). All of the modeling inputs used for modeling the 3D geometry of fault zones are described using continuous data types.

Simulation of scalar data is straightforward and well-established through the use of Markov-chain Monte Carlo (MCMC) sampling algorithms, easily accessible through the open source Python package *PyMC3* (Salvatier et al., 2016). The *PyMC3*

library has been demonstrated as a platform for performing MCUP of 3D geologic models (de la Varga and Wellmann, 2016; Schneeberger et al., 2017), and has even been implemented in the open source geologic modeling platform GemPy (de la Varga et al., 2019). An additional consideration in the case of continuous data types is the distinction between scalar and vectorial data (e.g., structural orientations). A probability distribution describing orientation data resides on the surface of a unit-sphere in 3D, and can be characterized using spherical probability distributions (Fisher et al., 1987; Mardia and Jupp, 2000). The benefit of

using spherical probability distributions to describe structural orientation uncertainty in 3D geologic modeling is clearly stated by Pakyuz-Charrier et al. (2018b), and their application in MCUP formulations continues to develop (Pakyuz-Charrier et al., 2018b, a; Carmichael and Ailleres, 2016). To remain concise, the following section focuses on the new contributions made to the use of spherical probability distributions utilizing the *R-fast* open source package available in the R language (Papadakis et al., 2018).

### 3.1 Spherical probability distributions

Fault orientations are vectors described by dip and dip azimuth components. Stereographic projection is often used to describe the fault plane using its pole (i.e., normal), defined by a unit vector or a trend and plunge. For orientation data in MCUP formulations, distributions of the Fisher-Bingham family (Bingham, 1964; Kent, 1982) provide a wide variety of choices for modeling varying degrees of uncertainty. Pakyuz-Charrier et al. (2018b) recently showed that scalar distributions are inadequate

for modeling the uncertainty of structural orientation data, providing an example using the von-Mises Fisher (vMF) distribution (spherical analogue to the isotropic bivariate normal distribution). This research continues the exploration into the use of





spherical distributions in MCUP by implementing the more general Bingham and Kent distributions (Fisher et al., 1987) from the Fisher-Bingham family to characterize anisotropic uncertainty of structural orientations used in 3D geologic modeling.

In structural geology, the use of spherical distributions to understand the uncertainty about structural orientation measure-
ments is well established (Mardia, 1981; Cheeney, 1983; Davis and Titus, 2017; Roberts et al., 2019). The open source Orient and Stereonet softwares by Vollmer (2018) and Allmendinger (2015) provide uncertainty estimates of structural data using spherical statistics, while Davis and Titus (2017) and Roberts et al. (2019) used spherical distributions to estimate confidence intervals about uncertain structural orientation measurements of folds and foliations. The analysis of anisotropic orientation uncertainty in structural geology is well established with Zhou and Maerz (2002), Peel et al. (2001), Carmichael and Ailleres
(2016) and Davis and Titus (2017) applying it to joint set identification, structural data clustering and foliation-lineation char-
acterization. While Pakyuz-Charrier et al. (2018a) showed clearly that the dip angle and dip azimuth should not be simulated independently as scalar values, it is apparent that the uncertainty affecting each of these aspects of the structural orientation need not be equal.

Two spherical distributions were reviewed in this study: the Bingham and Kent distributions. As opposed to the isotropic
vMF distribution, the Bingham and Kent distributions are capable of modeling potentially more realistic anisotropic uncertainty envelopes. This study explores a novel application of anisotropic spherical distributions in MCUP: characterizing subjective bias in the structural orientation uncertainty of fault zones (i.e., differing levels of information regarding the dip angle and dip azimuth of faults modeled in implicit 3D geologic models). The distributions, typically characterized from a series of input measurements, can also be characterized by directly controlling the input parameters themselves. This method allows for the
modeller to assume the size and shape of the structural orientation uncertainty envelope in the MCUP formulation.

The Bingham distribution is an antipodally symmetric distribution for axial data, defined explicitly in $R^3$ ($p = 3$) by a set of orthogonal eigenvectors ($\boldsymbol{e_1}, \boldsymbol{e_2}, \boldsymbol{e_3}$) and corresponding eigenvalues ($\lambda_1 \geq \lambda_2 \geq \lambda_3 = 0$). The eigenvector and eigenvalue pairs respectively detail the direction and degree of maximum, intermediate and minimum variance of the Bingham distribution. The distribution is described by Eq. 1 where $A = diag(\lambda_1, \lambda_2, \lambda_3)$ and $c(A)$ is the corresponding normalization constant (Fallaize
and Kypraios, 2016). Setting $\lambda_3 = 0$ merely ensures that the distribution shows maximum variance in the axial direction (i.e., across the unit sphere), allowing $\lambda_1$ and $\lambda_2$ to fully control the shape of the distribution when projected onto the lower hemisphere.

$$P(\boldsymbol{x}|A) = \exp\left(-\sum_{i=1}^{p-1}\lambda_i x_i^2\right)\frac{1}{c(A)} \qquad c(A) = \int_{\boldsymbol{x}\in S^{p-1}} \exp\left(-\sum_{i=1}^{p-1}\lambda_i x_i^2\right)dS^{p-1}(\boldsymbol{x}) \qquad (1)$$

The Kent distribution (or Fisher-Bingham 5-parameter distribution) is a more generalized form of the Bingham distribution,
defined for vectorial data focused around a known mean vector with an anisotropic uncertainty envelope. It is characterized by a mean vector ($\boldsymbol{x} = x_1, x_2, x_3$), the concentration parameter $\kappa$ and an ovalness parameter $\beta$. Its density is described by Eq. (2) for $\kappa \geq 0, \beta \geq 0$ where $\Omega$ is an orthogonal $p \times p$ matrix that can be likened to the eigenvector matrix used in simulating the Bingham distribution. The reader is referred to Appendix C of Pakyuz-Charrier et al. (2018a) for a more thorough explanation





of the parameterization of the Kent distribution.

$$P(\boldsymbol{x}|\Omega, \kappa\beta) = C_3(\kappa,\beta)\exp\left(\kappa\omega_1^T\boldsymbol{x} + \beta\big[(\omega_2^T\boldsymbol{x})^2 - (\omega_3^T\boldsymbol{x})^2\big]\right) \quad C_3(\kappa,\beta) = 2\pi\sum_{j}^{\infty}\frac{\Gamma\left(j+\frac{1}{2}\right)}{\Gamma(j+1)}\beta^{2j}\left(\frac{\kappa}{2}\right)^{-2j-\frac{1}{2}}I_{2j+\frac{1}{2}}(\kappa) \quad (2)$$

When modeling structural geologic orientation data, the distinction between axial (i.e., undirected) and vectorial (i.e., directed) data is irrelevant following the application of lower-hemisphere stereographic projection. The stereographic projection is applicable because in reality the orientation of geologic structures is defined by an axis laying within the plane of the structure. Regardless of the method of characterizing and sampling the orientation data, a stereographic projection to the
lower-hemisphere will return the conventional down-dip orientations as defined by dip/dip-azimuth or right-hand-rule systems.

### 3.2 Simulation

Simulating samples from spherical distributions requires dedicated algorithms separate from those used for scalar data types, due to the transformation between rectangular and spherical coordinates creating non-uniform areas of angular trend and plunge increments on the unit-sphere. Several solutions have been demonstrated to simulate random samples from spherical
distributions which are comprehensively documented in Kent et al. (2018). Simulation algorithms for distributions of the Fisher-Bingham family have been implemented by Papadakis et al. (2018) in an open-source R package *Rfast*. Open-source tools for statistical simulation in the R and Python environments (including their combined usage through the *rypy2* package), provide convenient, well-documented tools for applying established statistical techniques to novel fields in geoscience.

The algorithm for simulating random points from the Bingham and Kent distributions included in *Rfast* uses the acceptance-
rejection method, inspired by Kent et al. (2013) and Fallaize and Kypraios (2016). The method uses a Central Angular Gaussian (CAG) distribution as an envelope to approximate the Bingham distribution. The algorithm uses only the first two eigenvalues for identifiability, resulting in the need to use a rotation to align the sampled points with the desired orientation. The rotation of data sampled from spherical distributions to any new set of axes is possible due to the rotation-independence of the dispersion of spherical distributions.

### 210 3.3 Rotation

Two rotations using the Euler-Rodrigues formula (Dai, 2015) are useful for properly aligning the data simulated using the *Rfast* algorithms. The first, necessary for the Bingham distribution, interchanges two axes by a rotation of $\pi$ about an axis defined by the sum of the two axes to be interchanged, $\boldsymbol{k} = \boldsymbol{v_1} + \boldsymbol{v_2}$. The second rotation is necessary with either the Bingham or Kent distribution to correct for the arbitrary alignment of the orientation uncertainty envelope (Figure 3(a)), which is not desired
when characterizing anisotropic uncertainty of dip angle and dip azimuth of a fault. This occurs in the MCUP formulation when characterizing the distributions from input parameters directly rather than through eigen-decomposition or maximum likelihood estimation. An effective approach to rotating anisotropic spherical distributions was developed which rotates iteratively by small increments about the input orientation vector until an accuracy threshold on the desired alignment of the $\boldsymbol{e_3}$ plane is satisfied (Figure 3(b)).





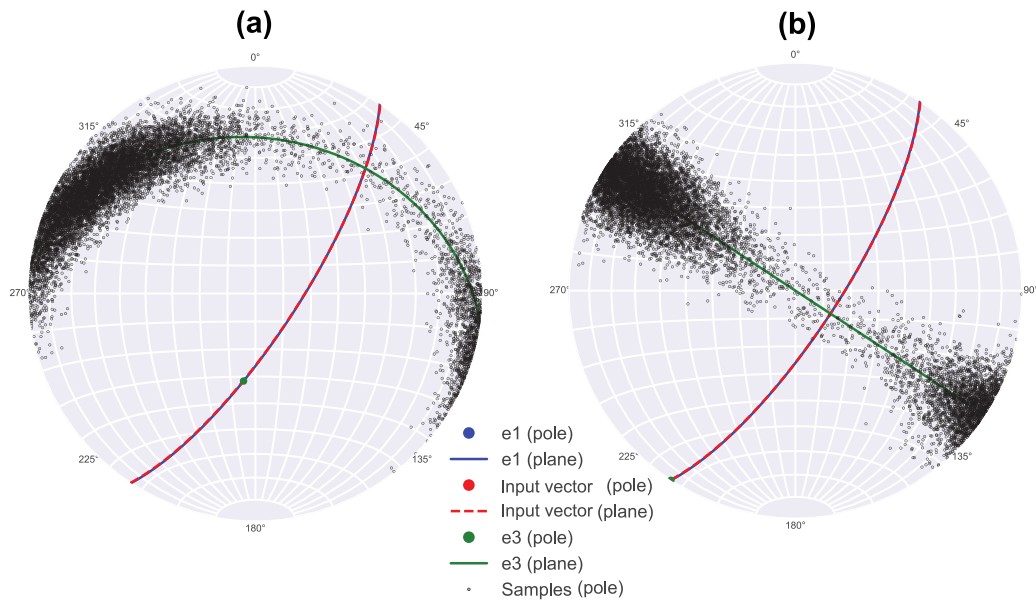

**Figure 3.** (a) Random $e_3$ orientation of anisotropic Bingham distribution resulting from simulation and (b) rotated distribution with properly aligned $e_3$ orientation.

## 4   Uncertainty assessment

After determining the appropriate probability distributions to use in the MCUP formulation, the uncertainty of chosen geologic modeling inputs is quantified based on geologic data and prior knowledge available. This essential step requires a thorough consideration of the types of uncertainty and methods for quantification available for each modeling input. To minimize potentially unwanted bias in the MCUP formulation, careful attention must be paid to understanding the geologic nature of the objective and subjective sources of uncertainty affecting each modeling input.

Two models were generated for this study from data derived from a historical geologic map in the Rocky Mountains of Colorado, USA (Figure 4). The first model contains a single fault zone (Figure 4(a)) and is analyzed to clearly illustrate the impacts of different geologic modeling inputs and uncertainty parameterizations on a 3D geologic model. The second model contains a network of major and minor fault zones (Figure 4(b)) and is provided to demonstrate how the methodology developed can be generalized to more complex models and to discuss the implications of fault zone interactions on geologic model uncertainty. The models follow the workflow illustrated in Figure 1 to simplify the 3D geometry of fault zones to satisfy the flexibility and automation required by MCUP.

The geology in the project area consists of uplifted, Precambrian crystalline igneous and metamorphic rocks. The most recent period of tectonic activity occurred from c. ~70-40 Ma. during the Laramide Orogeny which formed the modern day Rocky Mountains. During this time, brittle faulting occurred pervasively throughout the study area as part of the regional





Loveland-Berthoud Pass Fault Zone which passes just east of the study site, where it trends NNE for nearly 50 km (Lovering, 1935). A large number of fault zones of varying widths cross through the study area and were mapped at a 1:12,000 scale by Robinson et al. (1974). The geologic map (Figure 4) contains approximate traces and boundaries of fault zones in the study area, showing fault zones with thickness ranging from 5-50 ft (~1.5-15 m) as lines and fault zones greater than 50 ft (~15 m)

thick as hatched zones. Robinson et al. (1974) provided contoured stereonets of all faults mapped in the area, revealing that the majority of fault zones mapped dip steeply to the east/northeast. However, specific information on the dip angle of individual faults is missing from the geologic map. In addition to this significant source of orientation uncertainty, the historic map also includes significant geographical uncertainty resulting from georeferencing and drafting errors.

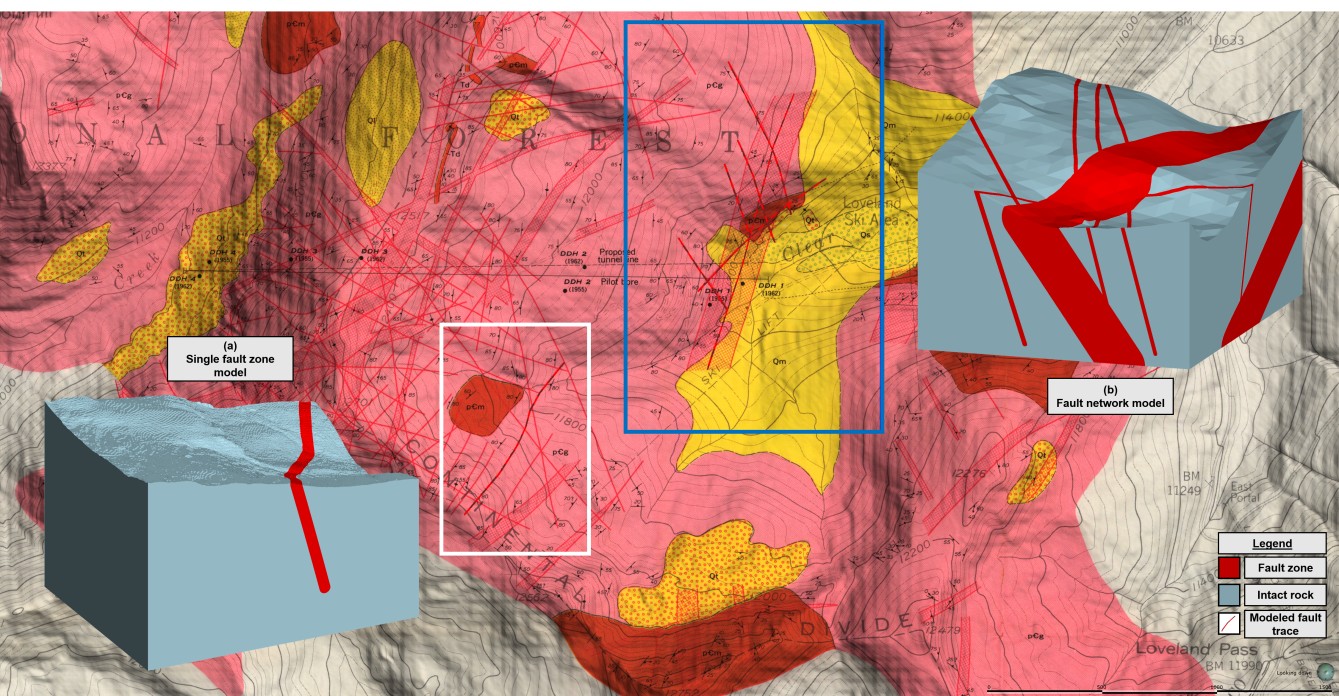

**Figure 4.** 1:12,000 geologic map from Robinson et al. (1974) showing mapped fault zones of varying widths. The white rectangle and associated overlay (a) show the single fault model while the blue rectangle and associated overlay (b) show the fault network model. Fault trace(s) used for modeling are highlighted within each rectangle.

Model realization creation is handled by custom Leapfrog Works back-end support developed for this study to allow for

automated updating of geologic modeling parameters from an initial model containing input fault zones. Model realizations generated in this manner are evaluated onto a grid of cells for subsequent analysis. The method put forward by Wellmann and Regenauer-Lieb (2012) is implemented to calculate the probability of occurrence of fault zone lithology in each cell. The probability of occurrence is then used to compute information entropy to describe the uncertainty of fault zones in the geologic model. In a binary system (e.g., fault zone vs. intact rock), information entropy is maximal when the probability of occurrence





of a fault zone is 50%, which as discussed in Krajnovich et al. (2020a) can introduce potential for misinterpretation of the geologic model uncertainty envelope if an inappropriate colormap is used.

A set of 1,000 realizations for each modeling input were propagated into the 3D geologic model independently and compared with the combined geologic model uncertainty assessment from all four modeling inputs (Figure 5). The significance of orientation and vertical termination depth uncertainty on the combined model uncertainty is clearly apparent at depth, while

the uncertainty of the fault zone near the ground surface is dominated by surface trace uncertainty. Uncertainty about the fault zone thickness appears to be largely overshadowed by that of the surface trace, which is considered to be a consequence of the significant georeferencing and drafting errors arising from the use of the historic geologic map (Section 6.1).

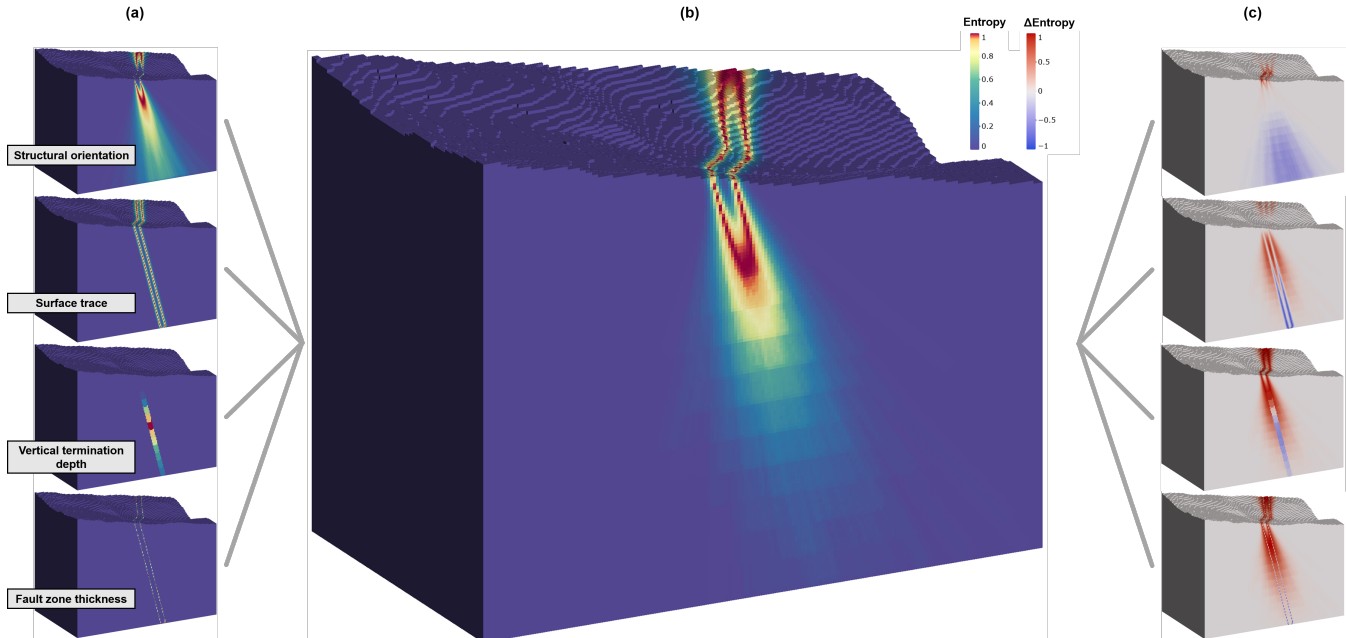

**Figure 5.** Block models showing information entropy quantified from (a) independent modeling inputs and (b) combined modeling inputs. The difference between the combined geologic model uncertainty and each independent modeling input is shown in (c), where blue values indicate that the independent modeling input showed greater entropy than the combined model uncertainty.

The following sections provide a detailed description of the methods implemented for parameterizing each geologic modeling input propagated into the model shown in Figure 5. The input perturbation script, which is compatible with virtually any

3D geologic modeling software, has been developed using open source code of the R and Python languages and is published at the code and data supplement. The interested reader is suggested to refer to the script alongside Sections 4.1-4.4 for exact results, figures and parameters of the MCUP formulation.





## 4.1 Structural orientation

The structural orientation of a fault zone varies along its surface and is implemented in modeling as a single dip azimuth and
dip angle vector applied to the implicit modeling interpolant. Natural variability of the fault surface and error of individual
measurements contribute objective uncertainty to the orientation used for modeling (Whitmeyer et al., 2019; Stigsson, 2016).
Additional, subjective sources of uncertainty are present that may affect the fault orientation in non-random ways (Bond, 2015),
including the application of prior knowledge (e.g., regional structural analysis) or measurement bias from difficulty interpreting
fault slip surfaces.

Stigsson (2016) shows that the objective uncertainty of measurement imprecision often underestimates the true uncertainty
associated with a geologic structure's orientation, suggesting that a more robust characterization of structural uncertainty
should consider potential sources of subjective bias. In the example of modeling fault zones from a geologic map (Figure 4),
anisotropic uncertainty untold by any single measurement is present due to the inconsistent information available regarding
fault zone dip azimuth and dip angle. An anisotropic Bingham distribution is simulated to capture the uncertainty of the input
orientation data from the geologic map. Observed variability in the surface trace and the regional structural analysis provided
in Robinson et al. (1974) were used to establish the uncertainty space used in simulation. Section 6.3 continues the discussion
regarding the use of an anisotropic structural orientation uncertainty envelope.

## 4.2 Surface trace

The surface trace of a fault zone is a polyline along the topography which follows an approximate central fault surface. In
ideal conditions, the fault trace would follow the centerline of the fault zone and reach along the whole length of the fault,
ending at the fault tip points. However, in reality, the centerline of the fault zone can rarely be determined exactly (Childs et al.,
2009). Additionally, when extracting the surface trace from a scanned geologic map, digitization error and geographical errors
are likely to be present as well. In the proposed MCUP formulation, the fault centerline inherits uncertainty from its possible
position within the perceived fault zone as well as from the relevant geographical errors.

The uncertainty affecting the surface fault trace results in changes in the trace location and shape. Independent perturbations
of the trace's endpoints are applied and linearly propagated along the fault trace to arrive at a smoothly varied location and
shape. A normal distribution characterizing the uncertainty about the location of each trace endpoint is parameterized from
the joint uncertainty stemming from fault zone centerline definition, digitization error and geographical errors in addition to a
random direction of shift obtained from a uniform distribution.

The three sources of uncertainty are quantified using the available information listed in respective order: average fault zone
thickness, published metrological studies (Zhong-Zhong, 1995) and approximate geographical error of known landmarks.
Additional details on the method of perturbing the location and shape of the surface trace based on the joint effect of these
sources of uncertainty are available in the code supplement.





### 4.3  Vertical termination depth

In implicit geologic modeling, by default, all faults extend through the entire model domain. To arrive at a more realistic approximation of the 3D fault geometry, a series of surfaces at fixed elevations are defined and used to terminate faults. The depth at which a fault is expected to terminate is based on prior knowledge obtained from past works into approximating the 3D geometry of faults (Walsh and Watterson, 1988; Nicol et al., 1996; Schultz and Fossen, 2001; Torabi et al., 2019a). These works established empirical relationships for fault surface geometry using an aspect ratio of fault length along strike

from tip to tip ($f_{length}$, henceforth length) vs. fault height along dip ($f_{height}$, henceforth height): $Aspect\ ratio = \frac{f_{length}}{f_{height}}$. The fault surface aspect ratio, while highly variable (ranging from ~1.5-16 in the sedimentary basin rocks studied by Torabi et al. (2019a)), has been demonstrated to be an effective measure for quantifying the 3D geometry of fault surfaces. While studies into the aspect ratio of faults in crystalline rock are scarce and typically focus on major thrust faults, a reasonable assumption is that the aspect ratio will lie in the lower range of that for sedimentary basin rocks (Torabi et al., 2019a) due to the lack of

mechanical stratigraphy; an aspect ratio of 1 to 5 was assumed to be appropriate for the MCUP formulation in this study.

The vertical termination depth used in modeling inherits uncertainty from the aspect ratio and persistence of the fault trace used. Lacking specific information on the location of fault tip points, an assumption of length must be made using the persistence of a fault trace at the surface as a proxy for the fault length. However, the endpoints of the fault trace may not be the true tip points due to artificial truncation by overburden or lack of outcrop. Lacking a detailed study of fault tip points,

this uncertainty can be characterized by averaging the fault trace lengths of a number of faults of the same group (i.e., similar orientation).

The vertical termination depth is sampled from a deterministic distribution combining uncertainty about the aspect ratio, fault trace persistence and prior information of the fault elevation and average dip. Sampled vertical termination depths are continuous, and are subsequently discretized onto the pre-defined termination surfaces in the geologic model. The scale at

which termination surfaces are defined is determined based on the end user needs of the geologic model; 50 meter intervals were used in this study to balance illustrative quality with model processing time.

### 4.4  Fault zone thickness

In the ideal case, a detailed study of fault zone thickness at outcrop provides a measure of the average thickness and its variability. Given the thickness range of 5-50 ft (1.5-15 m) on fault zones mapped as lines in Figure 4, a conservative estimate

on fault zone thickness in the project area was characterized using a normal distribution.

While not directly applied in the model shown in Figure 5, when lacking direct observations, fault zone thickness can also be approximated based on prior knowledge of a fault's estimated displacement using an established displacement to thickness (D:T) relationship appropriate for the project's geology (Torabi et al., 2019b; Childs et al., 2009). This approach would inherit uncertainty from the subjective interpretation of the fault's historical displacement and the empirical derivation of the fault's

D:T relationship. With a measure of displacement, a power-fit relationship allows for approximating the fault zone thickness. The input perturbation script includes functions for exploring the use of the D:T relationship based on a linearized power law





function provided by Torabi et al. (2019b), with curve-fitting parameters $\log_{10}(b)$ and $m$ to approximate fault zone thickness ($f_t$) from displacement ($f_d$), and a modifier $f_{CoreVsZone}$ used to model different sections of the fault zone based on the work by Childs et al. (2009). Uncertainty arises in the parameters of the D:T power-fit relationship, $\log_{10}(b)$ and $m$ and the assumed
fault displacement.

### 4.5  Simulation quality assessment

The quality of probabilistic simulation relies on primarily the size of the uncertainty space, the simulation method and the number of samples drawn. For any simulation, the realizations generated can be plotted in the data space and visually examined for appropriate coverage and shape (termed a realization plot). For non-spherical data types, use of MCMC simulation
methods creates trace plots and posterior histograms, which provide an intuitive method for independently assessing the quality of simulation for each input. Uniformity of the trace plot and width of the 95% highest posterior density (HPD) indicate, respectively, the convergence of the Markov Chain sampling and the thorough exploration of the posterior uncertainty space. Figure 6 shows an example of the realization plot, trace plots and posterior histograms generated for the simulation of vertical termination depths from Section 4.3. This figure allowed for identifying a strong tailing behavior in the output realizations,
leading to a reparameterization discussed in Section 6.2.

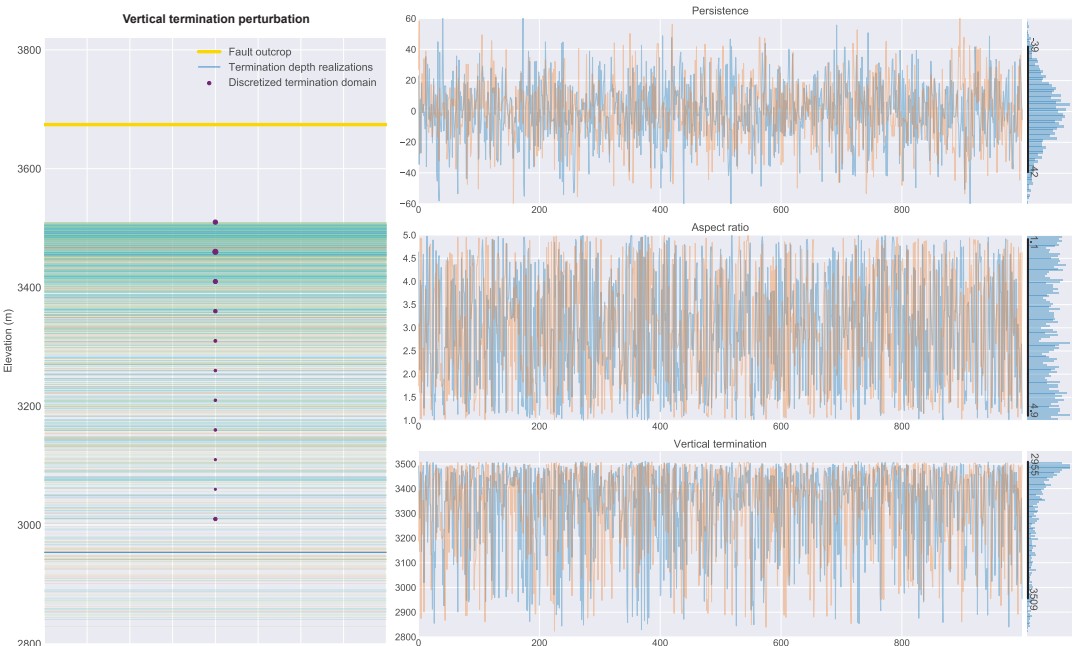

**Figure 6.** Realizations and Monte Carlo analysis results (trace plot and posterior histograms) from perturbation of the fault zone vertical termination depth based on a uniform distribution of fault aspect ratio. The 95% highest posterior density is overlain on the posterior histograms.





Trace plots are not available for the spherical data simulations due to reliance on the acceptance-rejection simulation method, while posterior histograms may be replaced by Exponential Kamb contouring Exponential Kamb contouring (Vollmer, 1995) or Rose diagrams to visualize the density of sampled poles across the surface of the unit sphere (as projected onto a lower-hemisphere projection). This visual assessment provides a semi-quantitative evaluation of the shape of the posterior spherical

probability distribution. Additionally, a recalculation of the eigenvector decomposition from the set of simulated samples provides a measure of the accuracy of the posterior distribution with respect to the input orientation values. Tools for generating figures for simulation quality assessment are provided and detailed in the input perturbation script.

Based on the assessment of simulation quality and consideration of compounding factors during uncertainty propagation, the MCUP formulation for the single fault model was run for a number of various realization counts (100, 300, 500, 1,000,

2,000 and 3,000). The processing time generally increases linearly with realization count, reaching many hours to several days for high realization counts on the single fault mock model containing 2.5 million cells. This study is intended to introduce and expand on the use of MCUP formulations for specific geologic modeling problems, and work regarding optimizing the efficiency of model processing is not a focus. The experiments conducted do highlight the need to understand (i) the realization requirement for exploring modeling inputs independently and its relationship to the size of the independent uncertainty spaces,

(ii) the interactions of various, related parameters during the uncertainty propagation step and (iii) identification of a balance between final model resolution, coverage, complexity and processing time.

## 5   Fault network model

The fault network model shown in Figure 4(b) contains a major fault zone (thickness = ~175 m) with five minor fault zones (thickness ≤ 16 m) branching out of it. The parameterization of modeling inputs was conducted following the methods de-

scribed in Section 4. The input orientations of each fault were determined based on the measured surface trace strike, and a dip angle assigned initially by a static random value based on the data published by Robinson et al. (1974). The resultant uncertainty model is shown in Figure 7, with cross-sections highlighting the intersection of three minor fault zones with the major fault zone. The modeling formulation scaled effectively to the more complex fault network model, which included two deterministic horizontal terminations of minor faults into the major fault zone.





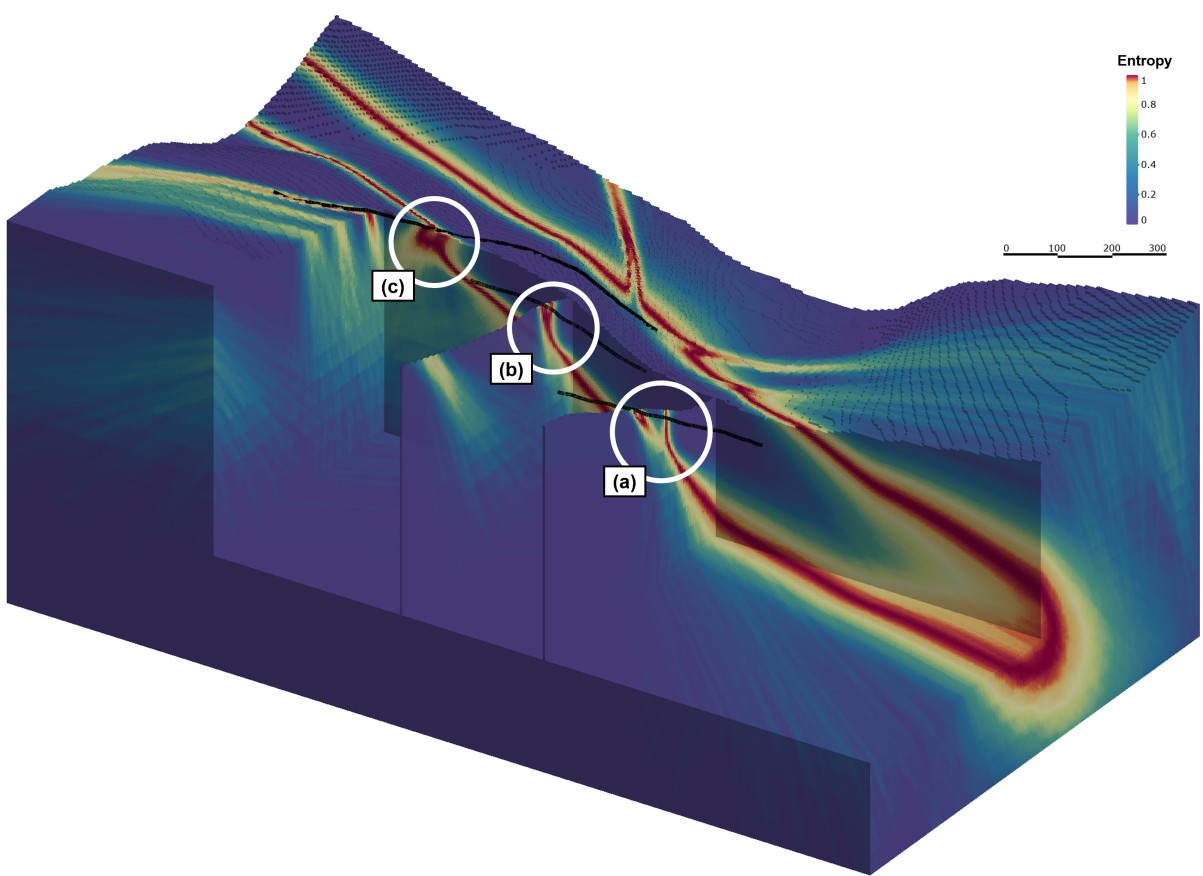

**Figure 7.** Combined geologic model entropy results for the fault network model, highlighting three intersection points between minor fault zones with the major fault zone: (a) approaching major fault zone boundary, (b) bordering major fault zone boundary and (c) overlapping boundaries.

Focusing on the intersection of fault zones reveals the interaction between fault zone uncertainty envelopes. In Figure 7, the slice (a) shows the uncertainty envelope of the major fault zone being deflected by a zone of lower entropy as the nearby minor fault zone approaches it. Slices (b) and (c) show the uncertainty envelopes of major and minor fault zones merging, showing how the deflection from (a) transfers into a hot spot of high uncertainty at the peak intersection point (overlapping boundaries). Just as the uncertainty of a single fault zone is maximal at its boundaries, the overlapping of two fault zone boundaries produces

an entropy hot spot.





## 6 Discussion

### 6.1 Historic dataset sensitivity

The observed sensitivity of the model uncertainty to the surface trace perturbation may be due to the relative significance of geographical sources of uncertainty in the case of modeling from a historic geologic map. These errors include inherent error
in the geologic map stemming from its accuracy and georeferencing errors arising from the conversion between geographical and projected coordinate systems. Georeferencing typically uses a rubber-sheet method, moving, rotating and stretching the map – in an ArcGIS platform – to optimally translate from the geographical coordinate system of the map (Lat/Long) to the projected coordinate system required by modeling (UTM Northing and Easting), and the error typically reported by these methods may underestimate the true error associated with the georeferenced map. Quantifying the true geographical error is
difficult, but can be approximated by comparing the location of known landmarks (e.g., intersection of roads, mountain tops, corners of buildings) between the georeferenced map and modern, digital datasets. From Figure 4, a maximum bound on the error in the location of known features across the geologic map was approximated to be 40 meters by visual examination of the georeferenced map. While the same methodology would apply to a surface trace obtained from modern mapping methods (e.g., global positioning systems), it is apparent that the contribution to model uncertainty could be drastically reduced. Reapplying
the methodology to a modern dataset may highlight different sensitivity.

### 6.2 Model reparameterization

Based on the inherent variability of fault aspect ratios, initially a bounded uniform distribution was determined to be appropriate (Figure 6(a)). However, the posterior distribution of vertical termination depths resulting from a bounded uniform parameterization of fault aspect ratio showed a strong tailing effect. This could be interpreted as being unrealistic, arising as an
artifact due to the shortening of vertical termination depth intervals for equivalent increases in fault aspect ratio. A reparameterization using a custom log-normal distribution of aspect ratio respecting similar maximum bounds of 1 and 5 is illustrated in Figure 8(b), showing a significant reduction in the tailing effect present in the vertical termination depth realizations. This reparameterization highlights the key strength – and susceptibility – of MCUP formulations, the reliance on a user defined characterization of input uncertainty. Again, it is necessary to reiterate that the modeler must take into consideration not only
field observations and theoretical prior knowledge when assessing a geologic modeling uncertainty formulation, but also their informed expectation of what is geologically realistic for their chosen modeling problem.





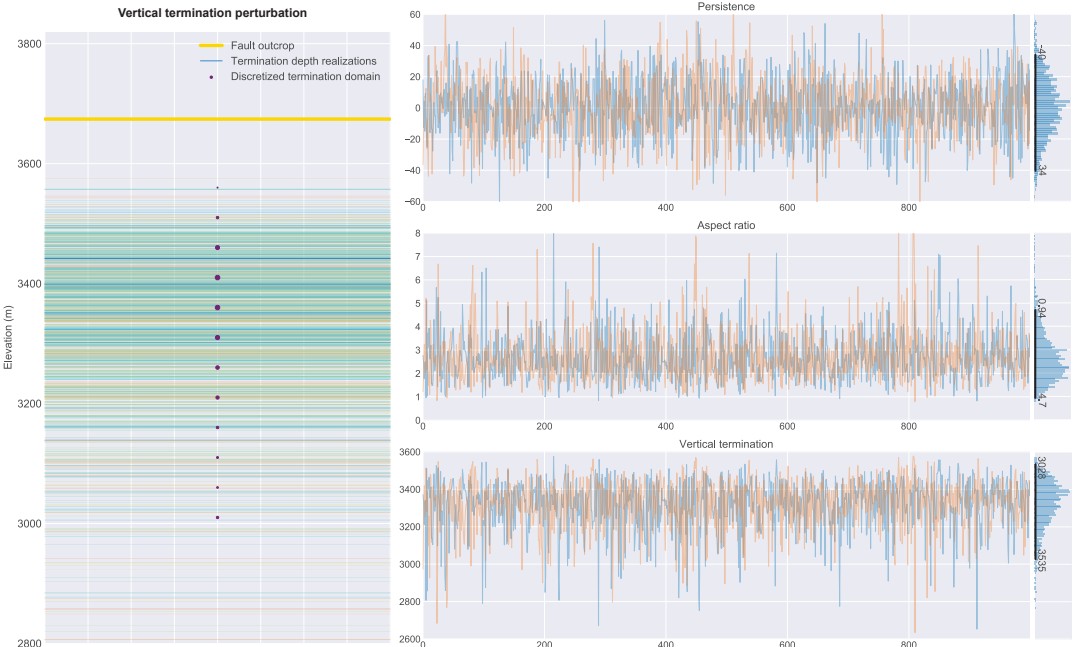

**Figure 8.** Realizations and Monte Carlo analysis results (trace plot and posterior histograms) from perturbation of the fault zone vertical termination depth, reparameterized using a log-normal distribution of fault aspect ratio. The 95% highest posterior density is overlain on the posterior histograms.

## 6.3 Parameter relationships

While many MCUP formulations – including this study – have focused on independent perturbations of geologic modeling inputs, relationships between modeling inputs are apparent in the modeling formulation developed. Intuitively, there is a re-
lationship between the average orientation of a fault's surface trace and the structural orientation applied to the fault surface modeling interpolant's global trend. Deviations in the modeled fault surface from the input orientations can occur when the two inputs are significantly different, typically arising in Leapfrog Works by way of overestimation of fault surface dip by up to 10 degrees when the surface trace azimuth (i.e., average normal to the fault surface trace) and global trend dip azimuth differed by greater than $20^o$. Comparing two geologic models generated with orientations sampled from anisotropic and isotropic
Bingham distributions with equivalent maximum uncertainty ranges (Figure 9) showed artifacts present in the geologic model uncertainty envelope when generated from the isotropic Bingham distribution. This issue is alleviated when the structural orientation is parameterized with an anisotropic Bingham distribution, allowing for increased variability in the dip angle without compromising the certainty of the dip azimuth.



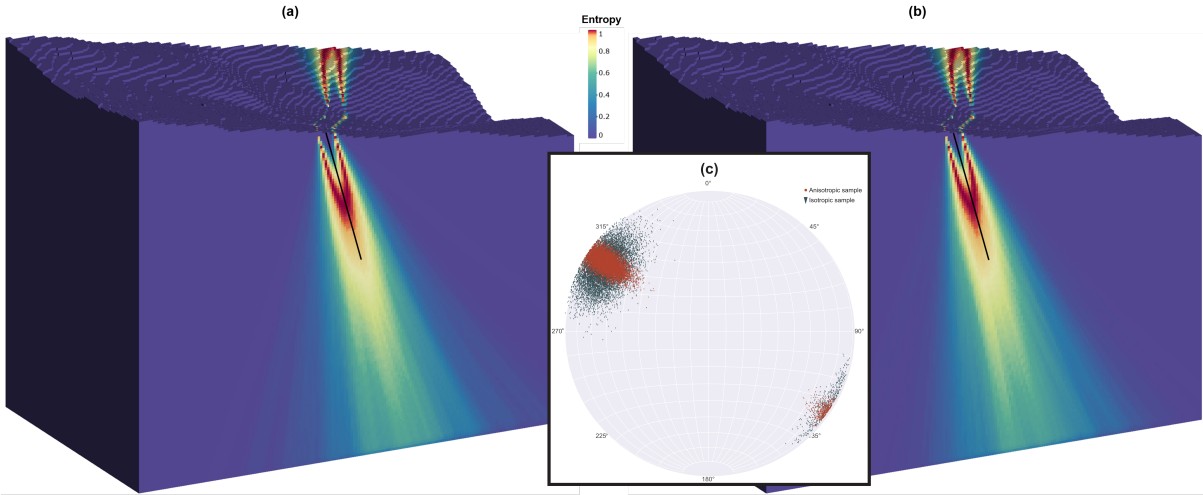

**Figure 9.** Block models generated from 1,000 realizations of (a) an anisotropic Bingham distribution and (b) an isotropic Bingham distribution. (c) Comparison of 10,000 orientation samples collected from anisotropic and isotropic Bingham distributions covering the same range of maximum uncertainty ($\lambda_2 = 22.5$).

The observed interaction between the surface trace and structural orientation inputs to the implicit 3D geologic model
suggests that coupling and correlation may be present between different geologic modeling inputs.

Relationships between modeling inputs also arise in different ways, for example the vertical termination depth and structural orientation uncertainty envelopes overlap heavily in the combined model uncertainty (Figure 5) leading to undersampling of the model uncertainty space when the independent uncertainty envelopes are combined. Similar behavior is observed when comparing orientation perturbations to fault zone thickness where thinner fault zones require finer orientation perturbations to
fully populate the uncertainty space of the 3D geologic model.

Coupling of the surface trace and structural orientation or structural orientation and vertical termination depth is not implemented in the current formulation due to inconsistencies in the sampling methods which required independent sampling of spherical and non-spherical distributions. Increasing the number of model realizations mitigated some of the effects of input correlations, though this comes at the expense of increased processing time. A treatment of these relationships through the
use of Gibbs sampling or other conditional methods of Monte Carlo simulation could potentially generate more realistic and efficient assessments of model uncertainty. However, until standard methods for simulating spherical and scalar data types in a single system are made available, creative model parameterizations – such as constraining the dip azimuth uncertainty to observed surface trace variability using an anisotropic spherical distribution – can circumvent some of the issues associated with these relationships.





### 6.4 Anisotropic spherical distributions

For modeling anisotropic uncertainty of structural orientation data, the Bingham and Kent distributions generate practically equivalent uncertainty envelopes, differing primarily in their parameterization. For the Bingham distribution, variations in the magnitudes of $\lambda_1$ and $\lambda_2$ allow for independently varying the size of the uncertainty space in orthogonal directions while for the Kent distribution variations in the values of $\kappa$ and $\beta$ can generate distributions with different overall size and ellipticity. To achieve independent uncertainty ranges of the dip angle and dip azimuth, both distributions require a series of rotations (described in Section 3.3) to properly align the distribution such that the major ellipse axis is aligned with a great circle with $90^o$ dip (Figure 3). Once aligned, the range of the Bingham distribution in the dip azimuth and dip angle directions can easily be varied independently through direct changes in the magnitudes of $\lambda_1$ and $\lambda_2$. The Kent distribution, however, introduced difficulty in varying these uncertainties independently due to coupling of the $\kappa$ and $\beta$ parameters. Compared to the Kent distribution, the Bingham distribution was observed to be more efficient at modeling strongly girdle-shaped distributions, which as discussed in Section 4.1 can be particularly useful for the limited data available from a geologic map. For these reasons, the Bingham distribution was chosen for the analysis in this study, though methods for simulating and rotating the Kent distribution are still provided for thoroughness.

### 6.5 Additional complexity for fault zone geometry

The author acknowledges that in reality fault zone geometry includes horizontal terminations. The mechanisms affecting the location of horizontal terminations are numerous and varied, including fault zone anastomosing, abrupt termination in intact rock and false trace termination due to obscuring by overburden. While several works have investigated the nature of fault-fault terminations using stochastic simulation (Aydin and Caers, 2017; Cherpeau and Caumon, 2015), due to the presence of other poorly defined sources of uncertainty, the placement of horizontal terminations in this study remained deterministic. Future work supplementing the limited dataset used in this study with detailed outcrop studies will be required for defining the nature and uncertainty of horizontal fault zone terminations.

Aside from horizontal fault terminations, fault zones present other interesting complexities that introduce additional levels of refinement for the developed modeling workflow. For example, internal fault zone composition is heterogeneous and modeling the different fault zone components (fault core, transition zone and damage zone) and could be implemented using the developed MCUP formulation if desired. Asymmetry of fault zone structure between the hanging wall and footwall has also been documented by Choi et al. (2016), which would require a new method of defining the distance function for generating the 3D fault zone volume. Similarly, variations in fault zone thickness along the area of the central fault surface are realistic, though would similarly require a new method for defining the distance function. These research questions enter into the realm of implicit geologic modeling theory, and are introduced merely to shed light on where refinements in model creation can benefit the modeling of fault zone structure.





# 7 Conclusions

The flexible MCUP method should continue to be leveraged to model novel problems in geologic modeling, such as the uncertainty of fault zones in 3D geologic models based on limited data from a historic geologic map and available prior knowledge. MCUP formulations developed should make full use of open source statistical packages in the R and Python

languages, many of which are experiencing ongoing development at the time of writing. While a unique modeling case was presented, the rationale applied to assessing uncertainty of poorly-constrained, preliminary geologic models sheds light on new ways to implement varied types of information in MCUP formulations.

Practical guidelines for developing MCUP formulations are also provided, reinforcing (i) the importance of developing a clear yet robust modeling workflow for the structure of interest, (ii) consideration of varied sources of geologic uncertainty

and (iii) creatively exploring the methods available for characterizing both objective and subjective modeling inputs. Prior knowledge in the form of established empirical and theoretical relationships present an opportunity to quantify and parameterize geologic modeling inputs that are usually interpretive, allowing for their inclusion in automated MCUP formulations. The Bingham distribution, while only moderately impactful on model uncertainty when comparing anisotropic and isotropic parameterizations, is recommended to replace the vMF distribution for modeling structural orientations due to the increased

flexibility of its parameterization. While the Bingham distribution was preferred in this study, the use of the Kent distribution appears to be practically equivalent.

Future work stemming from this preliminary modeling formulation may include incorporating new information in a Bayesian inference scheme to further refine the geologic model. Additionally, refining the modeling of fault zone internal structure and variability is recommended not only to further the usefulness of 3D geologic models in practical applications (e.g., subsurface

construction, fluid flow), but to also expand the understanding of the geometry and characteristics of these complex geologic structures.

*Code and data availability.* The necessary code and data for generating realizations of the geologic modeling inputs (including example results) is available at: https://github.com/ajkran2/Geologic-Model-Input-Uncertainty-Characterization (Krajnovich et al., 2020b).

*Author contributions.* Ashton Krajnovich is the primary researcher on this project responsible for the literature review, data acquisition,

geologic modeling, code development and writing. Dr. Wendy Zhou is the primary research advisor providing close supervision, feedback, guidance and revisions. Dr. Marte Gutierrez is the research co-advisor and is a primary contributor to forming the overarching research questions that this research attempts to address.

*Competing interests.* The author's declare that no competing interests are present.



*Acknowledgements.* This research is funded by the United States Department of Transportation University Transportation Center for Underground Transportation Infrastructure, Grant No. 69A3551747118. The authors would also like to gratefully acknowledge the outstanding support from Dr. Rose Pearson from Seequent for dedicating her time to develop automated generation of geologic model realizations and block model evaluations inside the Leapfrog Works software.



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
