# Peer review of "Uncertainty assessment for 3D geologic modeling of fault zones based on geologic inputs and prior knowledge"

_Solid Earth, 2020_

## Referee Comment (RC1) · Alexander Schaaf (Referee) · 14 Apr 2020

**0.1   General comments**

The paper presents a step forward in the uncertainty-aware modeling of subsurface faults in structural geomodels. The authors make use of Monte Carlo simulations to simulate uncertainty of fault zones based on a specific fault zone parameterization (surface traces, vertical termination surfaces, structural orientation and fault zone thickness) using a proprietary software suite. The authors elaborate the use of anisotropic spherical distributions for parameterizing orientation data for uncertainty simulation,

which is a valuable contribution.

The manuscript is overall well structured, except for a few re-arrangements necessary to increase readability (detailed in the specific comments). The authors give proper credit to related work and clearly indicate their own contribution. The title clearly reflects the contents of the paper. The figures presented will require some work to improve legibility and to avoid confusion of the reader.

But the authors appear to be confusing their simulation approach: They introduce MCUP (i.e. Monte Carlo simulation) in the methodology and properly parameterize their stochastic geomodel using probability distributions. But they then erroneously describe that they use Markov Chain Monte Carlo (MCMC) sampling. MCMC sampling is used for exploring the *posterior* space, which does not exist in a Monte Carlo simulation (i.e. MCUP). As the probability space is known in a Monte Carlo simulation, it needs no exploration. In a Monte Carlo simulation we only don't know how the combination of samples effect the output of the simulator function (the geomodeling software), thus we randomly sample (Monte Carlo sampling) from the parameter distributions to create a geomodel ensemble that shows us how the uncertainty in the input parameters effects the geomodel output. Luckily, to my knowledge, the used probabilistic programming framework *pymc3* defaults to Monte Carlo sampling when no likelihood function is given (and thus no Bayesian inference can be conducted). Thus the authors appear to have accidentally conducted the simulations they wanted to do (MCUP/Monte Carlo sampling). The use of trace plots (as in Figure 6 and 8) for Monte Carlo simulation results is meaningless though (and potentially misleading), as no sampler is being used that requires determination of convergence. As, luckily, the presented simulation results appear to be valid MCUP results, the authors only need to change their writing accordingly, without the need for re-running simulations.

In its current state, mixing up the terminology of Bayesian inference and MCUP, I can not recommend the paper to be accepted. But if the authors fix their method descriptions and discussions of the results to fit the MCUP simulations they actually conducted, I believe this could become a valid scientific contribution that is worth publishing in Solid Earth.

0.2   Specific comments

0.2.1   1 - Introduction

The Introduction of the paper needs to clearly state the scope of the study / hypothesis to be tested or explored.

0.2.2   2 - Model implementation

L52 - Both paragraph (lines 52-68) need to incorporated into the introduction as they define the scope and motivation of the study.

0.2.3   3 - Probability distributions for MCUP

L138 - How do you evaluate the likelihood of the proposal step in a Markov Chain during MCUP? This is only possible when doing a Bayesian inference, not a Monte Carlo uncertainty propagation, as you don't have any likelihood function.

Overall the description of simulation/sampling should be moved into Section 3.2.

L140 - The paper de la Varga & Wellmann (2016) uses *pymc* to conduct a Bayesian inference - thus not MCUP.

L141 - *pymc3* has not been implemented into *GemPy*, but rather *GemPy* in implemented in *theano*, which is also used in *pymc3*. Thus *GemPy* integrates seamlessly with *pymc3*, providing the gradients necessary for advanced gradient-based sampling

techniques such as Hamiltonian Monte Carlo.

**0.2.4  3.2 - Simulation**

A more adequate name for the section would be "Sampling".

**0.2.5  3.3 - Rotation**

This section is part of sampling and should be merged into Section 3.2

**0.2.6  4.2 - Surface trace**

L291 - It is unclear to me what the "approximate geographical error of known landmarks" is.

**0.2.7  4.3 - Vertical termination depth**

L312 - What is a deterministic distribution? Do you mean a derived distribution? Or an empirically parametrized distribution? A distribution should be by definitiv nondeterministic.

**0.2.8  4.5 - Simulation quality assessment**

L334 - Without a likelihood function you can't use a MCMC sampler, as you are unable to evaluate the step proposals.

L336 - In an MCUP simulation, you have no posterior uncertainty space, as you are not using any likelihood function. You are mixing up terminology of MCUP and Bayesian

inference. Again, MCMC sampling is only possible with a likelihood function (thus not in MCUP).

**0.2.9   6.2 - Model parametrization**

L388 - The use of "posterior distribution" is false, as you are doing MCUP, not a Bayesian inference.

**0.2.10   6.3 - Parameter relationships**

L411 - The meaning of the entire paragraph is unclear to me and needs to be revised.

L419 - Gibbs sampling is not applicable to MCUP, as no likelihood is used.

**0.2.11   Figures**

**0.2.12   Figure 2**

The dotted volume texture makes annotations for uncertainty extremely hard to read. The same goes for there fault zone signature/texture. I'd highly recommend removing as much texture as possible from the plot to improve legibility.

"The Visual Display of Quantitative Information" by Edward Tufte provides ample of additional reasons for reducing distracting "ink" from scientific visualizations and is well worth a read :-)

**0.2.13 Figure 4**

Highlighting of fault traces is really difficult to see. I highly recommend making this figure more legible to the reader by removing visual complexity: e.g. remove coloring of geological map in the background.

**0.2.14 Figure 5**

- legend is barely legible - please increase text size

- entropy plot of the fault zone thickness barely shows any uncertainty. If your discretization is not fine enough to resolve the simulated uncertainties, then is it worth incorporating into you model?

**0.2.15 Figure 6**

The use of trace plots is only useful if evaluating convergence of (e.g.) Markov chains. MCUP uses Monte Carlo simulation, thus the use of trace plots serves no purpose and is confusing. Also the rug plot on the left size shows the same information as the histogram of the vertical termination depth in the lower right. I'd recommend just using the histogram to demonstrate that you've sampled enough samples.

**0.2.16 Figure 8**

Same as for Figure 6.

---

## Referee Comment (RC2) · Florian Wellmann (Referee) · 24 Apr 2020

In the article "Uncertainty assessment for 3D geologic modeling of fault zones based on geologic inputs and prior knowledge", the authors present an application of a probabilistic geological modelling approach to assess uncertainties in fault zone models.

The work fits into the active field of probabilistic geological modelling and uncertainty assessment in 3-D structural models. The main contribution is the detailed consideration of different fault zone parameters, combined with the evaluation of different types of spherical distributions. In addition, the method is tested in a case study to investigate fault zone uncertainty in a Precambrian crystalline setting with two focus areas: one

investigating a single fault zone, and another model of a fault network. Both examples are well chosen to test the application in a realistic setting.

Interesting is specifically the description of sources of uncertainty for different faults zone parameters. Although some of the choices are clearly debatable (e.g. the fault aspect ratio), the authors state the problem of a lack of sufficient information - and this also implicitly highlights the relevance to, at least, consider these sources of uncertainty in a generated geological model, as done here.

The document is supported with code (written in Python and R), including a fully reproducible example for input file generation. The code is hosted on GitHub, and substituted with a license, ensuring the possibility for future use and adaptation. A brief suggestion: it would be good to provide a requirements file and/or more detailed installation instructions (using conda environments or a docker container), as the packages rpy2 and pymc3 are not part of common python distributions. The current version, used to generate results in this manuscript, are furthermore stored in a snapshot on Zeonodo, with a DOI.

One critical aspect in the manuscript is the description of the probabilistic model itself. There are several aspects that need to be adjusted (or clarified) before the manuscript can be considered for publication:

- The authors seem to mix MC draws from prior distributions (and the representation in models, i.e. the prior predictive models) with a full Bayesian inference. The tool that is employed, pymc, is fully capable of complete Bayesian inference methods, but as I see it (and I looked carefully, even in the code), there is no Bayesian model used here - even if the authors mention "likelihood" (but, I think, mixing terminology here a bit, see comments below) and "MCMC" in line 334. To my best interpretation, this is not the case here - if I am wrong, then please describe more clearly. Otherwise, the entire description on the probabilistic model needs a careful revision.

- The statistical parameters of the model are only broadly described, and the reader is referred to the online code. Not every interested reader can be expected to go through a submitted python code to understand and evaluate the scientific details and I strongly suggest to include details about the distributions (including hyperparamters) here in the text. Especially the deterministic transformation for fault depth termination would have been relevant, as it would have made a mistake more obvious: the (as I see it) wrong interpretation of the "tailing behaviour" in figure 6.

- The entire description in section 4.5 "Simulation quality assessment" needs to be revised: you can not present draws as traces, if they are not from a (sequential) MCMC chain, i.e.: you do not explore the posterior space and, therefore, you do not draw samples from a posterior distribution (if my interpretation is correct, see above) - the samples you obtain are simply draws from the (independent) prior distributions.

- Combining the previous two points: the interpretation of the tailing behaviour is, as I see it, wrong. It is not an effect of a posterior, but an effect of a derived distribution (or, in pymc lingo, a deterministic model combining multiple stochastic variables). This is quite obvious in your code (InputUncertaintyQuantification.py, lines 495ff):

```
""" Define the uniform or log-normal aspect ratio distribution """
if aspect_logn == 1:
    aspect_dist = pm.Lognormal('Aspect ratio', mu = term_mu, sd = term_sd)
else:
    aspect_dist = pm.Uniform('Aspect ratio',lower=aspect_min,upper=aspect_max

""" Define the distribution for variability of persistence """
persistence_dist = pm.Normal('Persistence', mu = 0, sd = persistence_std)
```

```
""" Deterministic distribution to compute termination depth based on
    above distributions """
zterm_dist = pm.Deterministic('Vertical termination',zo-np.sin(dip*pi/180)*(p
```

You do not sample from the posterior distribution, but combine two stochastic variables with a deterministic function. This is a very different concept (and completely fine, but needs adjustment in description).

A note on terminology: I know that "MCUP" has been used in some manuscripts in descent years. However, the concept itself (i.e. the sampling part, Monte Carlo sampling from a distribution and observing the propagation of uncertainty) is not new at all and in widespread use since the mid 20th century. I don't think it is particularly helpful to use a new term for a well established approach, and, on the contrary, will lead to unnecessary confusion for all researchers who are looking at the content and who are not familiar with the term - as it, at first, pretends to be something novel. I am confident that all appearances of MCUP and easily be reformatted without any loss of information and without making the document significantly longer:

- I would suggest to refer to the approach itself as "probabilistic modeling" (or "probabilistic geomodeling", when in the clear context of the geomodeling). All references to previous work can be kept, because this is what all the previous works also do.

- Instead of "MCUP formulation", simply refer to the setup or definition of the probabilistic model, where stochastic variables are defined by distributions (with corresponding hyperparameters). Adjusting the manuscript in this way will make the content a lot more accessible and comparable to other approaches, also outside the field of geomodeling itself.

More details on the workflow are also required. The provided python scripts are an important foundation, but they are only capable of producing the input data set for the geological model. A "Leapfrog back-end support" is mentioned in line 244. Is this method also available for other researchers? It is surely the case that the provided input script can be used to generate input data sets for many types of modeling approaches (even though surely not for all, line 259), but it hides a bit the fact that the forward modeling methods (which are generally far more complex) are often implemented in commercial software and often not accessible.

Comments to specific sections in the manuscript (identified by line number):

- 14: Note that the presented results are not a sensitivity analysis, but more a (visual) comparison of propagated uncertainties. A sensitivity analysis typically relates to a quantity of interest.

- 23: Would be suitable to include here the reference Wellmann  Caumon (2018) (not fishing for citations here, but as you reference it anyway later...).

- 38: See comments above for terminology (MCUP)

- 63: I am quite sure that there are studies on uncertainties in fault zone characterization. The topic of uncertainties about the characterization of fault zones has been discussed recently in a paper by Shipton et al. (2019, Fault Fictions: Cognitive biases in the conceptualization of fault zones.). But you probably refer to the treatment in a probabilistic geomodel? Please specify.

- 67: Here, for example, no need to refer exclusively to MCUP - this is true for any automated modeling approach.

- 78: RBFs are not really a rivaling approach - mostly another form for the kernel, otherwise a lot of similarity. Also, the first paper on RBF's in geomodeling dates back to 2002 (Cowan, on which Leapfrog is based), so also not "so" recent.

- 105: Suggestion: avoid "prior knowledge" when referring to likelihood functions, as these are different concepts. "Additional geological knowledge or observations" would make more sense.

- Fig. 2 caption: here, you actually consider different parameters of the probabilistic model (fault zone thickness, vertical termination, etc.) - not the sources of uncertainty? Please clarify or adjust.

- 118: You can only accommodate information in a form that can be included as a stochastic variable into the specific interpolation approach (see also treatment in Wellmann Caumon, 2018). It is possible to include additional information in the form of likelihood functions (e.g. de la Varga Wellmann, 2016), but this requires a full integration of modeling and a formulation in a Bayesian framework.

- 138: See above about terminology: you do not use MCMC here, so also referring to it here should be removed (even if the statement is true). If you want to mention MC, then simply MC sampling algorithms will do.

- 312: I don't understand this sentence here, as the same is true for all other continuous interfaces in the model, which are finally mapped onto a discrete mesh. Can be removed, in my opinion. Section 4.5: the entire section has to be removed or adjusted, as a posterior analysis does not make sense (see above), even if arviz produces nice plots. You can, of course, show samples of the model realizations (left side) and it is also fine to discuss the histogram to discuss the effect of the derived distribution. But any notion of posterior analysis and the plot of the chains are misleading.

- 350: Even for this high resolution, this seems like a long computation time. Can you comment on the relative timing for sampling (i.e. your script) and the runtime of the model using Leapfrog?

- 388: It is not a posterior distribution, but a derived distribution.

- 393: Also: typically correlations between parameters are unknown. Effect can (partly) be mitigated through an implementation in a full Bayesian framework with appropriate likelihood functions.

- 405: Which artefacts do you refer to? Please highlight or show difference plot, nothing too obvious.

- 420: Gibbs sampling is a variant of an MCMC approach, again: should not be referred to here. Do you simply mean appropriate sampling schemes from a joint distribution? Same difficulty concerning correlations as in line 393.

- 440: Authors, I assume?

- 472: The difficulty here: you would need to implement the forward model completely into the modeling framework (if information on the basis of the generated model should be considered). Is this possible with the approach used here (running Leapfrog as an integrated part of the Python script)? Please clarify.

Figures: overall good quality, but labels are generally far too small. Block diagrams also need orientation (at least North arrow) and axes labels should also be included.

---

## Author Comment (AC1) · 21 May 2020

The authors truly appreciate the detailed clarifying comments, especially regarding the description of the simulation approach. The authors acknowledge that there was confusion regarding how the simulation approach was described, primarily in the erroneous usage of the term Markov Chain Monte Carlo (MCMC) when the method applied in the study is in fact simply Monte Carlo sampling to explore the set of input probability distributions. The authors wish to reaffirm that the methodology employed in the study is intended to be that which the reviewer identified: the use of Monte Carlo sampling to explore the prior uncertainty space of geologic modeling

inputs. As is, the analysis performed in the study does not contain any use of Bayesian inference via MCMC (i.e., no likelihood functions were defined or used), nor was this an intended description of the methodology employed. The authors have taken steps throughout the text to remove any erroneous descriptions of the methodology and clarify the intended use of Monte Carlo sampling. A broad overview of the changes made to rectify this error include (i) removal of any mention of MCMC or posterior distributions and replacement with appropriate terminology and (ii) the removal of trace plots and a rethinking of the simulation quality assessment section.

Please refer to the attached supplement (.pdf) for the full, detailed replies to all comments of RC1 (general and specific) and the proposed changes to the manuscript.

Please also note the supplement to this comment:
https://www.solid-earth-discuss.net/se-2020-21/se-2020-21-AC1-supplement.pdf

─────────────────────────

[Figure]

**Supplement:**

**1 General comments**

*The paper presents a step forward in the uncertainty-aware modeling of subsurface faults in structural geomodels. The authors make use of Monte Carlo simulations to simulate uncertainty of fault zones based on a specific fault zone parameterization (surface traces, vertical termination surfaces, structural orientation and fault zone thickness) using a proprietary software suite. The authors elaborate the use of anisotropic spherical distributions for parameterizing orientation data for uncertainty simulation, which is a valuable contribution. The manuscript is overall well structured, except for a few re-arrangements necessary to increase readability (detailed in the specific comments). The authors give proper credit to related work and clearly indicate their own contribution. The title clearly reflects the contents of the paper. The figures presented will require some work to improve legibility and to avoid confusion of the reader.*

*But the authors appear to be confusing their simulation approach: They introduce MCUP (i.e. Monte Carlo simulation) in the methodology and properly parameterize their stochastic geomodel using probability distributions. But they then erroneously describe that they use Markov Chain Monte Carlo (MCMC) sampling. MCMC sampling is used for exploring the posterior space, which does not exist in a Monte Carlo simulation (i.e. MCUP). As the probability space is known in a Monte Carlo simulation, it needs no exploration. In a Monte Carlo simulation we only don't know how the combination of samples effect the output of the simulator function (the geomodeling software), thus we randomly sample (Monte Carlo sampling) from the parameter distributions to create a geomodel ensemble that shows us how the uncertainty in the input parameters effects the geomodel output. Luckily, to my knowledge, the used probabilistic programming framework pymc3 defaults to Monte Carlo sampling when no likelihood function is given (and thus no Bayesian inference can be conducted). Thus the authors appear to have accidentally conducted the simulations they wanted to do (MCUP/Monte Carlo sampling). The use of trace plots (as in Figure 6 and 8) for Monte Carlo simulation results is meaningless though (and potentially misleading), as no sampler is being used that requires determination of convergence. As, luckily, the presented simulation results appear to be valid MCUP results, the authors only need to change their writing accordingly, without the need for re-running simulations.*

*In its current state, mixing up the terminology of Bayesian inference and MCUP, I can not recommend the paper to be accepted. But if the authors fix their method descriptions and discussions of the results to fit the MCUP simulations they actually conducted, I believe this could become a valid scientific contribution that is worth publishing in Solid Earth.*

**Authors' answer:**

The authors truly appreciate the detailed clarifying comments, especially regarding the description of the simulation approach. The authors acknowledge that there was confusion regarding how the simulation approach was described, primarily in the erroneous usage of the term Markov Chain Monte Carlo (MCMC) when the method applied in the study is in fact simply Monte Carlo sampling to explore the set of input probability distributions. The authors wish to reaffirm that the methodology employed in the study is intended to be that which the reviewer identified: the use of Monte Carlo sampling to explore the prior uncertainty space of geologic modeling inputs. As is, the analysis performed in the study does not contain any use of Bayesian inference via MCMC (i.e., no likelihood functions were defined or used), nor was this an intended description of the methodology employed. The authors have taken steps throughout the text to remove any erroneous descriptions of the methodology and clarify the intended use of Monte Carlo sampling. A broad overview of the changes made to rectify this error include (i) removal of any mention of MCMC or posterior distributions and replacement with appropriate terminology and (ii)

the removal of trace plots and a rethinking of the simulation quality assessment section.

**2 Specific comments**

**2.1 1 - Introduction**

**2.1.1 *The Introduction of the paper needs to clearly state the scope of the study / hypothesis to be tested or explored.**

**Authors' answer:**

The two paragraphs previously in lines 52-68 have been incorporated fluidly into the introduction to specify the scope and motivation of the study.

**Changes to the manuscript:**

**Text moved to the end of Section 1 – Introduction. . .** This study expands the use of  probabilistic geomodeling to a new aspect of geologic modeling – fault zones, or the localized volume of fractured and displaced rock surrounding a finite fault surface, typically composed of a fault core and a damage zone (Caine et al., 1996; Childs et al., 2009; Peacock et al., 2016; Choi et al., 2016). Fault zones introduce regions of altered geotechnical strength and hydraulic permeability into the surrounding in-tact rockmass and are therefore of major importance to geological engineering projects that rely on accurate assessments of subsurface rock properties (e.g., tunnels, mines). While faults have been the focus of a significant amount of recent geologic modeling research (Røe et al., 2014; Cherpeau et al., 2010; Cherpeau and Caumon, 2015; Aydin and Caers, 2017), these works have focused on modeling fault surfaces directly rather than modeling the 3D geometry of fault zones. Detailed modeling of the 3D geometry of fault zones can improve the understanding of faults' impacts on geotechnical and reservoir engineering projects due to the fact that variations in fault zone thickness or composition can greatly alter the mechanical and hydrological behavior of a fault, e.g., its sealing potential (Caine et al., 1996; Fredman et al., 2008; Manzocchi et al., 2010).  Building on the existing literature on understanding the uncertainties about faults in the subsurface (Choi et al., 2016; Shipton et al., 2019; Torabi et al., 2019b), this study develops a novel, dedicated approach to leveraging probabilistic geomodeling to characterize the uncertainty of fault zones using 3D geologic models.

Fault zones may be irregular in shape, creating complex geometries which are difficult to characterize quantitatively (Torabi et al., 2019a, b). Peacock et al. (2016) provide a detailed list of the various types of damage zones and intersecting fault networks that comprise the general term "fault zone".  The inherent complexity of fault zone structure makes their precise modeling intractable in an automated geologic modeling application, such as that required by probabilistic geomodeling. A simplified approach to modeling fault zones in 3D geologic models is developed in this study based on the key elements defining fault zone geometry at a practical level of detail.

**2.2 2 - Model implementation**

**2.2.1 *L52 - Both paragraph (lines 52-68) need to incorporated into the introduction as they define the scope and motivation of the study.**

**Authors' answer:**

See above comment regarding reorganizing the content of the Introduction.

**2.3 3 - Probability distributions for MCUP**

**2.3.1 *L138 - How do you evaluate the likelihood of the proposal step in a Markov Chain during MCUP? This is only possible when doing a Bayesian inference, not a Monte Carlo uncertainty propagation, as you don't have any likelihood function.**

**Authors' answer:**

The erroneous mention of MCMC sampling algorithms has been removed and edited to specify the use of Monte Carlo sampling algorithms. This eliminates the question of "How do you evaluate the likelihood of the proposal step in a Markov Chain during MCUP?" by clarifying that only Monte Carlo sampling is performed, and not Bayesian inference.

**Changes to the manuscript:**

 Simulation of scalar data is straightforward and well-established through the use of Monte Carlo sampling algorithms, easily accessible through the open source Python package *PyMC3* (Salvatier et al., 2016).

**2.3.2 *Overall the description of simulation/sampling should be moved into Section 3.2.**

**Authors' answer:**

The description of simulation and sampling has been incorporated into Section 3.2.

**Changes to the manuscript:**

The paragraph referenced has been moved to Section 3.2: Simulation of scalar data is straightforward and well-established through the use of Markov-chain Monte Carlo (MCMC) sampling algorithms, easily accessible through the open source Python package *PyMC3* (Salvatier et al., 2016). The *PyMC3* library has been demonstrated as a platform for performing MCUE of 3D geologic models (de la Varga and Wellmann, 2016; Schneeberger et al., 2017), and has even been implemented in the open source geologic modeling platform GemPy (de la Varga et al., 2019). An additional consideration in the case of continuous data types is the distinction between scalar and vectorial data (e.g., structural orientations). A probability distribution describing orientation data resides on the surface of a unit-sphere in 3D, and can be characterized using spherical probability distributions (Fisher et al., 1987; Mardia and Jupp, 2000). The benefit of using spherical probability distributions to describe structural orientation uncertainty in 3D geologic modeling is clearly stated by Pakyuz-Charrier et al. (2018b), and their application in MCUE formulations continues to develop (Pakyuz-Charrier et al., 2018b, a; Carmichael and Ailleres, 2016). To remain concise, the following section focuses on the new contributions made to the use of spherical probability distributions utilizing the *R-fast* open source package available in the R language (Papadakis et al., 2018).

**2.3.3 *L140 - The paper de la Varga & Wellmann (2016) uses pymc to conduct a Bayesian inference - thus not MCUP.**

**Authors' answer:**

While the referenced study in question includes the use of the MCUP approach for probabilistic geomodeling, the authors realize the lack of clarity due to the referenced paper's focus on *PyMC*'s capability to perform Bayesian inference. The sentence has been reworded and expanded to emphasize the capability of *PyMC3* to perform Monte Carlo sampling outside of a

130    Bayesian inference, fitting its application in the current study.

**Changes to the manuscript:**

The *PyMC3*
135     The *PyMC3*
library is designed to facilitate Bayesian inference using computational sampling algorithms, though the inclusion of likelihood
functions is not required thereby allowing for utilization of the package functions for Monte Carlo sampling alone. The use
of *PyMC3* has been demonstrated successfully in the context of 3D geologic modeling by de la Varga and Wellmann (2016);
Schneeberger et al. (2017), and its implementation in *Theano* has allowed for seamless integration with the open source geo-
140    logic modeling platform GemPy (de la Varga et al., 2019). This study focuses solely on the step of probabilistic geomodeling
based on 3D geologic modeling inputs, leveraging only the Monte Carlo sampling capabilities of *PyMC3*.

**2.4    3.2 - Simulation**

**2.4.1    *A more adequate name for the section would be "Sampling".**

145    **Authors' answer:**

The section name has been changed to Sampling.

**2.5    3.3 - Rotation**

**2.5.1    *This section is part of sampling and should be merged into Section 3.2**

150    **Authors' answer:**

The section has been appended to Section 3.2 - Sampling.

**2.6    4.2 - Surface trace**

**2.6.1    *L291 - It is unclear to me what the "approximate geographical error of known landmarks" is.**

155    **Authors' answer:**

The section containing this sentence has been revised and expanded to clearly define the sources of uncertainty affecting the
fault trace and the methods with which they are quantified - including the geographical error arising from the use of a historic
geologic map.

160

**Changes to the manuscript:**

The uncertainty affecting the surface fault trace results in changes in the trace location and shape. Independent perturbations
of the trace's endpoints are applied and linearly propagated along the fault trace to arrive at a smoothly varied location and
165    shape. ~~A normal distribution characterizing the uncertainty about the location of each trace endpoint is parameterized from
the joint uncertainty stemming from fault zone centerline definition, digitization error and geographical errors in addition to a
random direction of shift obtained from a uniform distribution.~~ The three primary sources of uncertainty are quantified using the
available information listed in respective order: average fault zone thickness, published metrological studies (Zhong-Zhong,
1995) and approximate geographical error of known landmarks (e.g., mountain tops).

170

A bounded uniform distribution is parameterized to simulate a random direction of perturbation for each trace endpoint due to geographical error (i.e., drafting and georeferencing error). The normal distribution representing the total bound on geographical error is converted to respective $\hat{x}$ and $\hat{y}$ components using the directional cosine of the angle sampled from the

175 uniform distribution. This conversion to unit components is used similarly with the fault zone centerline definition uncertainty and digitization uncertainty using the acute angle $\theta$ between the orientation of the fault trace with the northing and easting directions. An additional logical check for the strike quadrant of the surface trace is required to implement this approach.

The individual sources of uncertainty affecting the surface trace endpoint locations are combined into a derived distribution

180 using a deterministic function to determine the total uncertainty affecting the location of each endpoint, given by Eq. 1.

$$P(\hat{x}|\sigma_{centerline}, \sigma_{dig}, \sigma_{geo}, \theta) = cos(\theta)\Big(N(0, \sigma_{centerline}) + N(0, \sigma_{dig})\Big) + N(0, \sigma_{geo})sin\Big(U(0, 2\pi)\Big),$$

$$P(\hat{y}|\sigma_{centerline}, \sigma_{dig}, \sigma_{geo}, \theta) = sin(\theta)\Big(N(0, \sigma_{centerline}) + N(0, \sigma_{dig})\Big) + N(0, \sigma_{geo})cos\Big(U(0, 2\pi)\Big)$$

(1)

The average fault zone thickness was used to characterize the fault zone centerline definition uncertainty affecting each surface trace endpoint. The geographical error was calculated to be approximately 40 meters based on the average distance

185 measured between known landmarks (e.g., mountain tops) on the geologic map and modern satellite imagery data. For both of these sources of uncertainty, the maximum error range described is treated as a 95% confidence interval, allowing a normal distribution to be parameterized with a mean of zero and a standard deviation equal to $maximum error/3.92$. The digitization error for a 1:12,000 map was represented by a normal distribution with a standard deviation of 3.666 m based on (Zhong-Zhong, 1995).

190 ## 2.7   4.3 - Vertical termination depth

**2.7.1   *L312 - What is a deterministic distribution? Do you mean a derived distribution? Or an empirically parametrized distribution? A distribution should be by definitiv nondeterministic.**

**Authors' answer:**

195 The phrase "deterministic distribution" was used following the *PyMC3* function terminology pm.Deterministic(), which is used for combining multiple stochastic variables (i.e., distributions) using a deterministic function. As the reviewer notes, this function is in fact empirically parameterized, and the more appropriate name "empirically derived distribution" – also known as a deterministic model combining multiple stochastic variables in *PyMC3* terminology – has been substituted. The description of the use of this style of distribution has been modified to clarify the reliance on combining several stochastic variables using

200 an empirically derived deterministic function.

**Changes to the manuscript:**

205  Sampling the uncertainty of the fault zone vertical termination depth involves combining multiple probability distributions using a deterministic function to generate an empirically derived probability distribution. In the derived distribution for fault zone vertical termination depth, $f_{length}$ and $Aspect ratio$ are characterized as independent probability distributions (respectively) and combined using a deterministic function based on the empirically derived description of 3D fault surface geometry, $z_{term} = z_{outcrop} - f_{height} * sin(\theta); f_{height} =$

210 $\frac{f_{length}}{Aspectratio}$. In this manner, the vertical termination depth ($z_{term}$) is calculated by converting the fault height to the vertical height using the average dip angle ($\theta$) and subtracting this from the average elevation of the fault outcrop ($z_{outcrop}$).

**2.8 4.5 - Simulation quality assessment**

**2.8.1 *L334 - Without a likelihood function you can't use a MCMC sampler, as you are unable to evaluate the step proposals.**

**2.8.2 *L336 - In an MCUP simulation, you have no posterior uncertainty space, as you are not using any likelihood function. You are mixing up terminology of MCUP and Bayesian inference. Again, MCMC sampling is only possible with a likelihood function (thus not in MCUP).**

**Authors' answer:**

The entire Section 4.5 has been revised in depth to remove erroneous inclusions of the terminology MCMC sampler and to remove the erroneous use of trace plots in assessing the quality of Monte Carlo simulation for exploring the prior uncertainty space.

**Changes to the manuscript:**

The quality of probabilistic simulation  is a product of the size of the uncertainty space, the simulation method used and the number of samples drawn. For any simulation, the realizations generated can be plotted in the data space and visually examined for appropriate coverage and shape (termed a realization plot). For  scalar data types,  histograms of the Monte Carlo draws provide an intuitive method for independently assessing the quality of simulation for each input.  Visual analysis of the shape of the histogram compared to the expected shape of the distribution and a comparison between the input distribution parameters (e.g., mean and standard deviation for a normal distribution) and their values calculated from the samples can quickly determine whether the samples drawn have sufficiently explored the uncertainty space. Figure 6 shows an example of the realization plot sample histograms generated for the simulation of vertical termination depths from Section 4.3. This figure allowed for identifying a strong tailing behavior in the output realizations, leading to a reparameterization discussed in Section 6.2.

[Figure]

**Figure 1.**  Visualization of Monte Carlo samples and associated geologic input realizations from perturbation of the fault zone vertical termination depth based on a uniform distribution of fault aspect ratio. The 95% highest  predictive density is overlain on the  histograms of the Monte Carlo samples.

240  For spherical data simulations, histograms may be replaced by Exponential Kamb contouring (Vollmer, 1995) or Rose diagrams to visualize the density of sampled poles across the surface of the unit sphere (as projected onto a lower-hemisphere projection). This visual assessment provides a semi-quantitative evaluation of the shape and distribution of the  sampled structural orientations. Additionally, a recalculation of the eigenvector decomposition from the set of simulated samples provides a measure of the accuracy of the posterior distribution with respect
245 to the input  parameter values. Tools for generating figures for simulation quality assessment are provided and detailed in the input perturbation script.

Based on the assessment of simulation quality and consideration of compounding factors during uncertainty propagation, the MCUE formulation for the single fault model was run for a number of various realization counts (100, 300, 500, 1,000, 2,000 and 3,000). The processing time generally increases linearly with realization count, reaching many hours to several days for
250 high realization counts on the single fault mock model containing 2.5 million cells. The vast majority of processing time is taken up by the model updating and block model calculation in Leapfrog. For the single fault mock model with 1,000 realizations and 2.5 million cells the sampling benchmark time was 87 seconds while the model processing benchmark time was 38.5 hours. This study is intended to introduce and expand on the use of MCUE formulations for specific geologic modeling problems, and work regarding optimizing the efficiency of model processing is not a focus. The experiments conducted do highlight the need
255 to understand (i) the realization requirement for exploring modeling inputs independently and its relationship to the size of the independent uncertainty spaces, (ii) the interactions of various, related parameters during the uncertainty propagation step and (iii) identification of a balance between final model resolution, coverage, complexity and processing time.

**2.9    6.2 - Model parameterization**

**2.9.1    *L388 - The use of "posterior distribution" is false, as you are doing MCUP, not a Bayesian inference.**

260 **Authors' answer:**

In line with the general comments regarding erroneous usage of terms from Bayesian inference (MCMC, posterior distributions), the description here has been revised to appropriately describe the prior predictive model that was explored using Monte Carlo sampling.

**Changes to the manuscript:**

However, the  empirically derived distribution of vertical termination depths resulting from a bounded uniform parameterization of fault aspect ratio showed a strong tailing effect (right skewed).

**2.10  6.3 - Parameter relationships**

**2.10.1  *L411 - The meaning of the entire paragraph is unclear to me and needs to be revised.**

**Authors' answer:**

The specified paragraph on addressing the observed relationships among geologic modeling input parameters using the MCUP formulation has been revised clarify the authors' stance on why and how these parameter relationships arise in the MCUP formulation. In essence, the paragraph is intended to highlight the potential for undersampling the geologic model uncertainty space when considering overlapping uncertainty envelopes of individual model inputs.

**Changes to the manuscript:**

~~Relationships between modeling inputs also arise in different ways, for example the vertical termination depth and structural orientation uncertainty envelopes overlap heavily in the combined model uncertainty (Figure 5) leading to undersampling of the model uncertainty space when the independent uncertainty envelopes are combined. Similar behavior is observed when comparing orientation perturbations to fault zone thickness where thinner fault zones require finer orientation perturbations to fully populate the uncertainty space of the 3D geologic model.~~ Despite performing a thorough exploration of each, independent parameter's uncertainty during Monte Carlo sampling (Section 4.5), undersampling of the combined geologic model uncertainty space can still occur during uncertainty propagation. An example of this arises when considering the vertical termination depth and structural orientation. Truncation of fault zone realizations at any given termination interval effectively reduces the number of realizations available for sampling the full range of structural orientation uncertainty at deeper intervals. This is evidenced in Figure 5(b) by the increasing prevalence of "stair-stepping" artefacts in the combined model uncertainty with depth.

**2.10.2  *L419 - Gibbs sampling is not applicable to MCUP, as no likelihood is used.**

**Authors' answer:**

The erroneous mention of Gibbs sampling has been removed. The section has been revised to better highlight the potential for exploring the input uncertainty space using joint distributions among various parameters believed to be correlated.

**Changes to the manuscript:**

A treatment of these relationships through  parameterizing previously independent input probability distributions using a joint distribution (and an appropriate sampling scheme) could potentially generate more realistic and efficient assessments of model uncertainty.

**3 Figures**

**3.1 Figure 2**

*The dotted volume texture makes annotations for uncertainty extremely hard to read. The same goes for there fault zone signature/texture. I'd highly recommend removing as much texture as possible from the plot to improve legibility.*

*"The Visual Display of Quantitative Information" by Edward Tufte provides ample of additional reasons for reducing distracting "ink" from scientific visualizations and is well worth a read :-)*

**Authors' answer:**

The suggested changes have been made to the fault zone schematic figure, clearing away non-informative ink and emphasizing the illustrated uncertainty envelopes. Thank you very much to the reviewer for the kind suggestion on better practices for graphical display.

**Changes to the manuscript:**

[Figure]

**Figure 2.** A schematic showing  possible uncertainty envelopes about the four geologic modeling inputs used to characterize the 3D geometry of a fault zone in the subsurface. Modified from Krajnovich et al. (2020).

**3.2 Figure 4**

*Highlighting of fault traces is really difficult to see. I highly recommend making this figure more legible to the reader by removing visual complexity: e.g. remove coloring of geological map in the background.*

**Authors' answer:**

325

 The figure has been edited to make the fault traces bolder and to provide a stronger contrast between the fault traces and the geological map in the background.

**Changes to the manuscript:**

330

[Figure]

**Figure 3.** 1:12,000 geologic map from Robinson et al. (1974) showing mapped fault zones of varying widths. The white rectangle and associated overlay (a) show the single fault model while the blue rectangle and associated overlay (b) show the fault network model. Fault trace(s) used for modeling are highlighted within each rectangle as green polylines.

**3.3   Figure 5**

-  *legend is barely legible - please increase text size*

-  *entropy plot of the fault zone thickness barely shows any uncertainty. If your discretization is not fine enough to resolve the simulated uncertainties, then is it worth incorporating into you model?*

335   **Authors' answer:**

 The legend text size has been increased to match the average text size of the paper's text.

 The reviewer's comment is interesting and insightful, and is addressed briefly in the preceding section. While it is quite 340   clear that the fault zone thickness uncertainty is largely insignificant in this single fault model (where the fault zone thickness was 8 m, vs. the 5 m block size), the authors observed that the wider fault zone present in the second, fault network model

would contribute more heavily to the model uncertainty. This reasoning, and the desire to present general recommendations to modeling the uncertainty of fault zones, led the authors to leave the results as is for the single fault model.

345 **Changes to the manuscript:**

[Figure]

**Figure 4.** Block models showing information entropy quantified from (a) independent modeling inputs and (b) combined modeling inputs. The difference between the combined geologic model uncertainty and each independent modeling input is shown in (c), where blue values indicate that the independent modeling input showed greater entropy than the combined model uncertainty.

**3.4  Figure 6**

*The use of trace plots is only useful if evaluating convergence of (e.g.) Markov chains. MCUP uses Monte Carlo simulation,*
*thus the use of trace plots serves no purpose and is confusing. Also the rug plot on the left size shows the same information as*
350 *the histogram of the vertical termination depth in the lower right. I'd recommend just using the histogram to demonstrate that*
*you've sampled enough samples.*

**Authors' answer:**

355   In line with this specific comment and the general comments above, the erroneous referrals to MCMC methods and the use of trace plots have been eliminated from the paper. They have been replaced by the appropriate discussion of assessing the exploration of the input uncertainty space using graphical representations of realizations and histograms of the Monte Carlo samples.

**Changes to the manuscript:**
360

[revised manuscript text omitted]

---

## Author Comment (AC2) · 21 May 2020

The authors truly appreciate the detailed constructive comments from the referee. The authors have taken several steps to address the misuse of terminology regarding Markov Chain Monte Carlo (MCMC) algorithms and posterior distributions. As the referee noted, there is and was no intention of a Bayesian model being used in the study. The manuscript has been edited throughout to replace erroneous mentions of terms related to MCMC and Bayesian inference (e.g., MCMC sampling, posterior distributions) to proper terminology for the Monte Carlo sampling that was performed in the study.

[Figure]

As for the description of the statistical parameters of the Monte Carlo model, sections of text that were previously withheld for the sake of brevity have been reintroduced to Sections 4.1-4.4 to describe in detail the parameterization of and rationale behind the distributions explored using Monte Carlo sampling.

Relating to the above point regarding erroneous usage of terms from Bayesian inference, Section 4.5 has been reworked extensively to provide an appropriate description of how the study went about assessing the quality of the exploration of the input uncertainty space from Monte Carlo sampling.

The authors agree that the interpretation of the tailing behavior is a result of the use of an empirically derived distribution (previously referred to as a "deterministic distribution"), and not a result of posterior analysis. The wording has been adjusted to clearly state this. However, the authors chose to retain the interpretation of tailing behavior in the vertical termination depth. The authors believe it highlights the possibility of unexpected uncertainty envelope shapes when using empirically derived probability distributions.

The authors have also updated the README file included with the code published on Github to provide a clearer and more comprehensive snapshot of the dependencies required for use of the input uncertainty quantification script.

The authors agree with the referee's well thought out recommendation regarding the extraneous use of abbreviations when referring to the concept of probabilistic geomodeling through exploration of the geologic model input uncertainty space using Monte Carlo sampling. Changes have been made throughout the paper to refer to

the method as 'probabilistic geomodeling', following a proper introduction of what the term means in the context of the study. This includes changing the term "MCUP formulation" to "probabilistic model".

The authors acknowledge that the description of the automated workflow implemented in Leapfrog with custom support is sparse, and have supplemented additional text to detail the process of automated model updating. The authors believe that the automated model updating that was implemented for this study is in fact rather straightforward, in the sense that it follows the same series of modeling steps that a user in Leapfrog would follow if they wished to create $n$ realizations of their own fault zone model. The authors believe that the updated text clearly illustrates this concept to the reader.

The method implemented in Leapfrog is available to other researchers on the basis that they contact the developers of Leapfrog (Seequent) independently to acquire access to the unique functionality (which is built on top of a default Leapfrog installation). The product is not currently commercially available and was designed with the supervision of the authors to accomplish the specific goals of the current study. The authors of this study are not developers of Leapfrog, and are therefore not privy to the specific code written in the Leapfrog development environment to accomplish the automated model updating. Rather, the authors worked in collaboration with the developers of Leapfrog to guide them in implementing our own requirements for automated model updating. Communication between the authors and the developers of Leapfrog provided a sufficient level of transparency in how the code was developed, although the specific code cannot be released to the public as it is built directly within the Leapfrog engine.

Please refer to the attached supplement (.pdf) for the full, detailed replies to all

comments of RC2 (general and specific) and the proposed changes to the manuscript.

Please also note the supplement to this comment:
https://www.solid-earth-discuss.net/se-2020-21/se-2020-21-AC2-supplement.pdf

————————————————————

[Figure]

**Supplement:**

**1 General comments**

*In the article "Uncertainty assessment for 3D geologic modeling of fault zones based on geologic inputs and prior knowledge", the authors present an application of a probabilistic geological modelling approach to assess uncertainties in fault zone models.*

*The work fits into the active field of probabilistic geological modelling and uncertainty assessment in 3-D structural models. The main contribution is the detailed consideration of different fault zone parameters, combined with the evaluation of different types of spherical distributions. In addition, the method is tested in a case study to investigate fault zone uncertainty in a Precambrian crystalline setting with two focus areas: one investigating a single fault zone, and another model of a fault network. Both examples are well chosen to test the application in a realistic setting.*

*Interesting is specifically the description of sources of uncertainty for different faults zone parameters. Although some of the choices are clearly debatable (e.g. the fault aspect ratio), the authors state the problem of a lack of sufficient information - and this also implicitly highlights the relevance to, at least, consider these sources of uncertainty in a generated geological model, as done here.*

*The document is supported with code (written in Python and R), including a fully reproducible example for input file generation. The code is hosted on GitHub, and substituted with a license, ensuring the possibility for future use and adaptation. A brief suggestion: it would be good to provide a requirements file and/or more detailed installation instructions (using conda environments or a docker container), as the packages rpy2 and pymc3 are not part of common python distributions. The current version, used to generate results in this manuscript, are furthermore stored in a snapshot on Zeonodo, with a DOI.*

*One critical aspect in the manuscript is the description of the probabilistic model itself. There are several aspects that need to be adjusted (or clarified) before the manuscript can be considered for publication:*

- *The authors seem to mix MC draws from prior distributions (and the representation in models, i.e. the prior predictive models) with a full Bayesian inference. The tool that is employed, pymc, is fully capable of complete Bayesian inference methods, but as I see it (and I looked carefully, even in the code), there is no Bayesian model used here - even if the authors mention "likelihood" (but, I think, mixing terminology here a bit, see comments below) and "MCMC" in line 334. To my best interpretation, this is not the case here - if I am wrong, then please describe more clearly. Otherwise, the entire description on the probabilistic model needs a careful revision.*

- *The statistical parameters of the model are only broadly described, and the reader is referred to the online code. Not every interested reader can be expected to go through a submitted python code to understand and evaluate the scientific details and I strongly suggest to include details about the distributions (including hyperparamters) here in the text. Especially the deterministic transformation for fault depth termination would have been relevant, as it would have made a mistake more obvious: the (as I see it) wrong interpretation of the "tailing behaviour" in figure 6.*

- *The entire description in section 4.5 "Simulation quality assessment" needs to be revised: you can not present draws as traces, if they are not from a (sequential) MCMC chain, i.e.: you do not explore the posterior space and, therefore, you do not draw samples from a posterior distribution (if my interpretation is correct, see above) - the samples you obtain are simply draws from the (independent) prior distributions.*

45      – *Combining the previous two points: the interpretation of the tailing behaviour is, as I see it, wrong. It is not an effect of a posterior, but an effect of a derived distribution (or, in pymc lingo, a deterministic model combining multiple stochastic variables). This is quite obvious in your code (InputUncertaintyQuantification.py, lines 495ff):*

```
""" Define the uniform or log-normal aspect ratio distribution """
if aspect_logn == 1:
    aspect_dist = pm.Lognormal('Aspect ratio', mu = term_mu, sd = term_sd)
else:
    aspect_dist = pm.Uniform('Aspect ratio',lower=aspect_min,upper=aspect_max)

""" Define the distribution for variability of persistence """
persistence_dist = pm.Normal('Persistence', mu = 0, sd = persistence_std)

""" Deterministic distribution to compute termination depth based on above distributions
zterm_dist = pm.Deterministic('Vertical termination',zo-np.sin(dip*pi/180)*(persistences
```

    *You do not sample from the posterior distribution, but combine two stochastic variables with a deterministic function.*
60     *This is a very different concept (and completely fine, but needs adjustment in description).*

**Authors' answer:**

    The authors truly appreciate the detailed constructive comments from the referee. The authors have taken several steps to address the misuse of terminology regarding Markov Chain Monte Carlo (MCMC) algorithms and posterior distributions. As
65 the referee noted, there is and was no intention of a Bayesian model being used in the study. The manuscript has been edited throughout to replace erroneous mentions of terms related to MCMC and Bayesian inference (e.g., MCMC sampling, posterior distributions) to proper terminology for the Monte Carlo sampling that was performed in the study.

    As for the description of the statistical parameters of the Monte Carlo model, sections of text that were previously withheld
70 for the sake of brevity have been reintroduced to Sections 4.1-4.4 to describe in detail the parameterization of and rationale behind the distributions explored using Monte Carlo sampling.

**Changes to the manuscript:**

[revised manuscript text omitted]

**. . . Text added to Section 4.4, Fault zone thickness:** For the sake of visualization in the single fault example, a normal distribution characterizing fault zone thickness used a conservative parameterization of $\mu = 30m$ and $\sigma = 2m$.

Relating to the above point regarding erroneous usage of terms from Bayesian inference, Section 4.5 has been reworked extensively to provide an appropriate description of how the study went about assessing the quality of the exploration of the input uncertainty space from Monte Carlo sampling.

The authors agree that the interpretation of the tailing behavior is a result of the use of an empirically derived distribution (previously referred to as a "deterministic distribution"), and not a result of posterior analysis. The wording has been adjusted to clearly state this. However, the authors chose to retain the interpretation of tailing behavior in the vertical termination depth. The authors believe it highlights the possibility of unexpected uncertainty envelope shapes when using empirically derived probability distributions.

The authors have also updated the README file included with the code published on Github to provide a clearer and more comprehensive snapshot of the dependencies required for use of the input uncertainty quantification script.

*A note on terminology: I know that "MCUP" has been used in some manuscripts in descent years. However, the concept itself (i.e. the sampling part, Monte Carlo sampling from a distribution and observing the propagation of uncertainty) is not new at all and in widespread use since the mid 20th century. I don't think it is particularly helpful to use a new term for a well established approach, and, on the contrary, will lead to unnecessary confusion for all researchers who are looking at the content and who*

*are not familiar with the term - as it, at first, pretends to be something novel. I am confident that all appearances of MCUP and easily be reformatted without any loss of information and without making the document significantly longer:*

- *I would suggest to refer to the approach itself as "probabilistic modeling" (or "probabilistic geomodeling", when in the clear context of the geomodeling). All references to previous work can be kept, because this is what all the previous works also do.*

- *Instead of "MCUP formulation", simply refer to the setup or definition of the probabilistic model, where stochastic variables are defined by distributions (with corresponding hyperparameters). Adjusting the manuscript in this way will make the content a lot more accessible and comparable to other approaches, also outside the field of geomodeling itself.*

**Authors' answer:**

The authors agree with the referee's well thought out recommendation regarding the extraneous use of abbreviations when referring to the concept of probabilistic geomodeling through exploration of the geologic model input uncertainty space using Monte Carlo sampling. Changes have been made throughout the paper to refer to the method as 'probabilistic geomodeling', following a proper introduction of what the term means in the context of the study. This includes changing the term "MCUP formulation" to "probabilistic model".

*More details on the workflow are also required. The provided python scripts are an important foundation, but they are only capable of producing the input data set for the geological model. A "Leapfrog back-end support" is mentioned in line 244. Is this method also available for other researchers? It is surely the case that the provided input script can be used to generate input data sets for many types of modeling approaches (even though surely not for all, line 259), but it hides a bit the fact that the forward modeling methods (which are generally far more complex) are often implemented in commercial software and often not accessible.*

**Authors' answer:**

The authors acknowledge that the description of the automated workflow implemented in Leapfrog with custom support is sparse, and have supplemented additional text to detail the process of automated model updating. The authors believe that the automated model updating that was implemented for this study is in fact rather straightforward, in the sense that it follows the same series of modeling steps that a user in Leapfrog would follow if they wished to create $n$ realizations of their own fault zone model. The authors believe that the updated text clearly illustrates this concept to the reader.

The method implemented in Leapfrog is available to other researchers on the basis that they contact the developers of Leapfrog (Seequent) independently to acquire access to the unique functionality (which is built on top of a default Leapfrog installation). The product is not currently commercially available and was designed with the supervision of the authors to accomplish the specific goals of the current study. The authors of this study are not developers of Leapfrog, and are therefore not privy to the specific code written in the Leapfrog development environment to accomplish the automated model updating. Rather, the authors worked in collaboration with the developers of Leapfrog to guide them in implementing our own requirements for automated model updating. Communication between the authors and the developers of Leapfrog provided a sufficient level of transparency in how the code was developed, although the specific code cannot be released to the public as it is built directly within the Leapfrog engine.

**Changes to the manuscript:**

Model realization creation is handled by custom Leapfrog Works back-end support developed for this study to allow for automated updating of geologic modeling parameters from an initial model containing input fault zones. The initial model must be created in Leapfrog using the workflow provided in Figure 1, and naming conventions for the geologic model, fault zones, polylines and termination surfaces being specified in a user-generated text file. The custom version of Leapfrog Works

uses this text file in conjunction with the Leapfrog model and input realization files to automatically step through creation
185 of each geologic model realization based on the simulated data, following the workflow from Figure 1. Model realizations generated in this manner are automatically evaluated onto a grid of cells defined by the text file for subsequent analysis. The method put forward by Wellmann and Regenauer-Lieb (2012) is implemented to calculate the probability of occurrence of fault zone lithology in each cell. The probability of occurrence is then used to compute information entropy to describe the uncertainty of fault zones in the geologic model. In a binary system (e.g., fault zone vs. intact rock), information entropy
190 is maximal when the probability of occurrence of a fault zone is 50%, which as discussed in Krajnovich et al. (2020) can introduce potential for misinterpretation of the geologic model uncertainty envelope if an inappropriate colormap is used. The method implemented in Leapfrog can be made available to other researchers on the basis that they contact the developers of Leapfrog (Seequent) independently to inquire about access to the unique functionality (which is built on top of a default Leapfrog installation).

**2 Specific comments**

**2.1 Line 14: *Note that the presented results are not a sensitivity analysis, but more a (visual) comparison of propagated uncertainties. A sensitivity analysis typically relates to a quantity of interest.**

**Authors' answer:**

200     The authors recognize that the term "sensitivity analysis" typically refers to a specific method of analysis performed on a probabilistic model, and the usage has been revised to match the "visual comparison of the contribution of independent input uncertainties to the geologic model uncertainty" performed in the study.

**Changes to the manuscript:**

205

    The MCUP formulation developed is applied to a simple geologic model built from historically available geologic mapping data  allowing for a visual comparison of the independent  contributions of each modeling input on the combined model uncertainty, revealing that vertical termination depth and structural orientation uncertainty dominate model uncertainty at depth while surface trace uncertainty dominates model uncertainty near the ground surface.

210

**2.2 Line 23: *Would be suitable to include here the reference Wellmann Caumon (2018) (not fishing for citations here, but as you reference it anyway later...).**

**Authors' answer:**

215     The reference to Wellmann and Caumon, 2018 has been added here as the authors agree that it is a fundamental work towards the growing use of 3D geologic models for prediction and communication of subsurface geology. Instead of Line 23, it was added in the reference of Line 22.

**Changes to the manuscript:**

220

     Three-dimensional (3D) geologic models are becoming the state of the art for the prediction and communication of subsurface geology in a wide range of projects (Turner and Gable, 2007; Wellmann and Caumon, 2018) including. . .

**2.3 Line 38: *See comments above for terminology (MCUP)**

**Authors' answer:**

Changes to the use of the "MCUP" and "MCUP formulation" terminology have been made throughout the paper in line with the general comment addressed above.

**Changes to the manuscript:**

 Recently, the well established Monte-Carlo simulation method has been adopted into a widely used method for 3D geologic model uncertainty assessment by way of uncertainty propagation of geologic model inputs into the 3D geologic model space (Wellmann and Caumon, 2018). The method, henceforth termed "probabilistic geomodeling", focuses on the impact of uncertainty in geologic modeling inputs on a 3D geologic model by generating a set of model realizations based on perturbations in selected modeling inputs, sampled using Monte Carlo simulation algorithms.  Probabilistic geomodeling is flexible, allowing for a wide variety of uncertainty sources affecting various geologic modeling inputs to be quantified by the user and propagated into the 3D geologic model.

**2.4 Line 63: *I am quite sure that there are studies on uncertainties in fault zone characterization. The topic of uncertainties about the characterization of fault zones has been discussed recently in a paper by Shipton et al. (2019, Fault Fictions: Cognitive biases in the conceptualization of fault zones.). But you probably refer to the treatment in a probabilistic geomodel? Please specify.**

**Authors' answer:**

The authors acknowledge the lack of clarity when referring to characterizing the uncertainties present in understanding fault zone structure. The section has been revised to specify the application of studied uncertainties about fault zone structure to a probabilistic geomodeling workflow.

**Changes to the manuscript:**

 Building on the existing literature on understanding the uncertainties about faults in the subsurface (e.g., Choi et al., 2016; Shipton et al., 2019; Torabi et al., 2019a), this study develops a novel, dedicated approach to leveraging probabilistic geomodeling to characterize the uncertainty of fault zones using 3D geologic models.

**2.5 Line 67: *Here, for example, no need to refer exclusively to MCUP - this is true for any automated modeling approach.**

**Authors' answer:**

The reference to complex modeling of fault zone structures has been revised to refer to any automated geologic modeling workflow, rather than only the previously mentioned MCUP approach.

**Changes to the manuscript:**

 The inherent complexity of fault zone structure makes their precise modeling intractable in an automated geologic modeling application, such as that required by probabilistic geomodeling.

270 **2.6 Line 78:** *RBFs are not really a rivaling approach - mostly another form for the kernel, otherwise a lot of similarity. Also, the first paper on RBF's in geomodeling dates back to 2002 (Cowan, on which Leapfrog is based), so also not "so" recent.*

**Authors' answer:**

275 The sentence intended to illustrate that RBF based implicit modeling approaches are comparable to the more common co-kriging based methods has been revised to properly respect the literature behind RBF based implicit modeling.

**Changes to the manuscript:**

280  Implicit geologic modeling using RBFs is comparable in quality to modeling using popular co-kriging approaches (Cowan et al., 2003; Hillier et al., 2014, 2017).

**2.7 Line 105:** *Suggestion: avoid "prior knowledge" when referring to likelihood functions, as these are different concepts. "Additional geological knowledge or observations" would make more sense.*

285 **Authors' answer:**

The sentence has been revised as suggested to avoid confusion between "prior knowledge" and likelihood functions in the context of a Bayesian model.

290 **Changes to the manuscript:**

The second method, operating within a Bayesian inference scheme, is to incorporate  additional geological knowledge or observations to validate – or rather, as Tarantola (2006) states, invalidate – model realizations.

295 **2.8 Fig. 2 caption:** *here, you actually consider different parameters of the probabilistic model (fault zone thickness, vertical termination, etc.) - not the sources of uncertainty? Please clarify or adjust.*

**Authors' answer:**

The sentence has been revised to match the figure's content, which is a depiction of possible uncertainty envelopes about the
300 four geologic inputs used in 3D modeling of fault zones.

**Changes to the manuscript:**

A schematic showing  possible uncertainty envelopes about the four geologic modeling inputs used
305 to characterize the 3D geometry of a fault zone in the subsurface. Modified from Krajnovich et al. (2020).

**2.9 Line 118:** *You can only accommodate information in a form that can be included as a stochastic variable into the specific interpolation approach (see also treatment in Wellmann Caumon, 2018). It is possible to include additional information in the form of likelihood functions (e.g. de la Varga Wellmann, 2016), but this requires a full integration of modeling and a formulation in a Bayesian framework.*

310 **Authors' answer:**

The authors agree with the referee's comment regarding what kind of information can be incorporated into the MCUP formulation. The sentence has been revised to highlight that any additional observations that can be accommodated into the modeling workflow are required to be of the same format as the four inputs considered in the study, i.e., inputs to the implicit modeling interpolation.

**Changes to the manuscript:**

While formulated for a case with limited data, the developed  probabilistic geomodeling approach allows for accommodating additional observations of the fault zone geologic modeling inputs (e.g., via a modern outcrop study)  through a straightforward reparameterization of the input probability distributions to include the new data.

**2.10    Line 138:** *See above about terminology: you do not use MCMC here, so also referring to it here should be removed (even if the statement is true). If you want to mention MC, then simply MC sampling algorithms will do.*

**Authors' answer:**

The erroneous mention of MCMC sampling algorithms has been removed and edited to specify the use of Monte Carlo sampling algorithms.

**Changes to the manuscript:**

 Simulation of scalar data is straightforward and well-established through the use of Monte Carlo sampling algorithms, easily accessible through the open source Python package *PyMC3* (Salvatier et al., 2016).

**2.11    Line 312:** *I don't understand this sentence here, as the same is true for all other continuous interfaces in the model, which are finally mapped onto a discrete mesh. Can be removed, in my opinion.*

**Authors' answer:**

The sentence has been revised to clarify the idea that the implementation of fault zone vertical terminations in the developed fault zone modeling approach relies on a series of pre-defined horizontal termination surfaces. Unlike the discretization of the implicit scalar field isovalues onto meshes (which is handled by the Leapfrog modeling engine), this step requires the user to pre-generate termination surfaces at specifically chosen depths in the subsurface prior to performing probabilistic geomodeling steps.

**Changes to the manuscript:**

Sampled vertical termination depths  are subsequently discretized onto the pre-defined termination surfaces  created during the initial geologic modeling step (Figure 1(b)). The  interval of these termination surfaces  can be adjusted based on the end user needs of the geologic model; 50 meter intervals were used in this study to balance illustrative quality with model processing time.

**2.12    Section 4.5:** *the entire section has to be removed or adjusted, as a posterior analysis does not make sense (see above), even if arviz produces nice plots. You can, of course, show samples of the model realizations (left side) and it is also fine to discuss the histogram to discuss the effect of the derived distribution. But any notion of posterior analysis and the plot of the chains are misleading.*

**Authors' answer:**

The entire Section 4.5 has been revised in depth to remove erroneous terminology of an MCMC sampler and to remove the erroneous use of trace plots in assessing the quality of Monte Carlo simulation for exploring the prior uncertainty space.

**Changes to the manuscript:**

The quality of probabilistic simulation  is a product of the size of the uncertainty space, the simulation method used and the number of samples drawn. For any simulation, the realizations generated can be plotted in the data space and visually examined for appropriate coverage and shape (termed a realization plot). For  scalar data types,  histograms of the Monte Carlo draws provide an intuitive method for independently assessing the quality of simulation for each input.  Visual analysis of the shape of the histogram compared to the expected shape of the distribution and a comparison between the input distribution parameters (e.g., mean and standard deviation for a normal distribution) and their values calculated from the samples can quickly determine whether the samples drawn have sufficiently explored the uncertainty space. Figure 1 shows an example of the realization plot sample histograms generated for the simulation of vertical termination depths from Section 4.3. This figure allowed for identifying a strong tailing behavior in the output realizations, leading to a reparameterization discussed in Section 6.2.

[Figure]

**Figure 1.**  Visualization of Monte Carlo samples and associated geologic input realizations from perturbation of the fault zone vertical termination depth based on a uniform distribution of fault aspect ratio. The 95% highest  predictive density is overlain on the  histograms of the Monte Carlo samples.

 For spherical data simulations, histograms may be replaced by Exponential Kamb contouring (Vollmer, 1995) or Rose diagrams to visualize the density of sampled poles across the surface of the unit sphere (as projected onto a lower-hemisphere projection). This visual assessment provides a semi-quantitative evaluation of the shape and distribution of the  sampled structural orientations. Additionally, a recalculation of the eigenvector decomposition from the set of simulated samples provides a measure of the accuracy of the posterior distribution with respect to the input  parameter values. Tools for generating figures for simulation quality assessment are provided and detailed in the input perturbation script.

Based on the assessment of simulation quality and consideration of compounding factors during uncertainty propagation, the MCUE formulation for the single fault model was run for a number of various realization counts (100, 300, 500, 1,000, 2,000 and 3,000). The processing time generally increases linearly with realization count, reaching many hours to several days for high realization counts on the single fault mock model containing 2.5 million cells. The vast majority of processing time is taken up by the model updating and block model calculation in Leapfrog. For the single fault mock model with 1,000 realizations and 2.5 million cells the sampling benchmark time was 87 seconds while the model processing benchmark time was 38.5 hours. This study is intended to introduce and expand on the use of MCUE formulations for specific geologic modeling problems, and work regarding optimizing the efficiency of model processing is not a focus. The experiments conducted do highlight the need to understand (i) the realization requirement for exploring modeling inputs independently and its relationship to the size of the independent uncertainty spaces, (ii) the interactions of various, related parameters during the uncertainty propagation step and (iii) identification of a balance between final model resolution, coverage, complexity and processing time.

**2.13 Line 350: *Even for this high resolution, this seems like a long computation time. Can you comment on the relative timing for sampling (i.e. your script) and the runtime of the model using Leapfrog?**

**Authors' answer:**

The runtime of the model generation in Leapfrog is approximately three orders of magnitude longer than the timing for sampling ( 38 hrs vs. 1.5 minutes). The focus of the study was admittedly on the input uncertainty quantification, and efficiency of the automated model updating process was not sought after at this stage. Benchmark statistics on an example run of sampling and model generation have been introduced into the text to clarify this.

**Changes to the manuscript:**

The processing time generally increases linearly with realization count, reaching many hours to several days for high realization counts on the single fault mock model containing 2.5 million cells. The vast majority of processing time is taken up by the model updating and block model calculation in Leapfrog. For the single fault mock model with 1,000 realizations and 2.5 million cells the sampling benchmark time was 87 seconds while the model processing benchmark time was 38.5 hours.

**2.14 Line 388: *It is not a posterior distribution, but a derived distribution.**

**Authors' answer:**

Correct, and the appropriate revision has been made in line with this specific comment and the earlier general comments.

**Changes to the manuscript:**

However, the  empirically derived distribution of vertical termination depths resulting from a bounded uniform parameterization of fault aspect ratio showed a strong tailing effect (right skewed).

420 **2.15 Line 393:** *Also: typically correlations between parameters are unknown. Effect can (partly) be mitigated through an implementation in a full Bayesian framework with appropriate likelihood functions.*

**Authors' answer:**

The authors agree with the referee's comments regarding lack of knowledge about the possible correlations between model-
425 ing parameters. The authors however believe that the suggested commentary fits naturally into the next Section 6.3 - Parameter relationships. The following sentence has been appended onto the end of Section 6.2 to smoothly transition into that discussion.

**Changes to the manuscript:**

430 This reparameterization highlights the key strength – and susceptibility – of MCUP formulations, the reliance on a user defined characterization of input uncertainty. Again, it is necessary to reiterate that the modeler must take into consideration not only field observations and theoretical prior knowledge when assessing a geologic modeling uncertainty formulation, but also their informed expectation of what is geologically realistic for their chosen modeling problem. Furthermore, reparame-terizing individual aspects of the probabilistic model may prove to be insufficient due to the presence of inherently unknown
435 relationships between the chosen model parameters.

**2.16 Line 405:** *Which artefacts do you refer to? Please highlight or show difference plot, nothing too obvious.*

**Authors' answer:**

The phrase artefacts has been replaced with the more accurate concept of a skewed uncertainty envelope. The figure in
440 question has also been replaced with a closer view of the uncertainty envelopes, highlighting the skewing (i.e., tendency of realizations towards more vertical dips) suspected due to disagreement between surface trace and structural orientation in-puts when sampling orientations from wide, isotropic spherical distributions. A difference plot is also included to highlight the slight skewing of the geologic model uncertainty envelope towards steeper dipping fault zones when parameterized using wide, isotropic spherical distributions.
445

**Changes to the manuscript:**

Deviations in the modeled fault surface from the input orientations can occur when the two inputs are significantly different, typically arising in Leapfrog Works by way of overestimation of fault surface dip by up to $10^o$ when the surface trace azimuth
450 (i.e., average normal to the fault surface trace) and global trend dip azimuth differed by greater than $20^o$ (Krajnovich et al., 2020). Comparing two geologic models generated with orientations sampled from anisotropic and isotropic Bingham distribu-tions with equivalent maximum uncertainty ranges (Figure 9) showed  a skewing of the  geologic model uncertainty envelope when generated from the isotropic Bingham distribution. The results show a consistent skew towards near-vertical dips of the modeled fault zone realizations. This issue is alleviated when the structural orientation is parameter-
455 ized with an anisotropic Bingham distribution, allowing for increased variability in the dip angle without compromising the certainty of the dip azimuth.

[Figure]

**Figure 2.** Visual comparison of geologic model entropy generated using anisotropic (a) and isotropic (b) Bingham distributions. The entropy difference between these two models (c) highlights slight skewing of the uncertainty envelope towards steeper dipping fault zones when parameterized using samples from an isotropic Bingham distribution (d). Entropy difference is shown such that negative values indicate higher entropy in the isotropic orientation model.

**2.17 Line 420: *Gibbs sampling is a variant of an MCMC approach, again: should not be referred to here. Do you simply mean appropriate sampling schemes from a joint distribution? Same difficulty concerning correlations as in line 393.**

**Authors' answer:**

The erroneous reference to Gibbs sampling has been removed, and the correction that the referee identified has been added to the section.

**Changes to the manuscript:**

A treatment of these relationships through  parameterizing previously independent input probability distributions using a joint distribution (and an appropriate sampling scheme) could potentially generate more realistic and efficient assessments of model uncertainty. While the correlation between parameters in a probabilistic model is typically unknown, in some cases – such as the correlation between fault trace azimuth and structural orientation azimuth – the correlation can be assumed based on available prior geologic knowledge of the parameter's real-world relationships.

**2.18 Line 440: *Authors, I assume?**

**Authors' answer:**

Correct.

**Changes to the manuscript:**

The authors acknowledges that in reality fault zone geometry includes horizontal terminations.

**2.19 Line 472:** *The difficulty here: you would need to implement the forward model completely into the modeling framework (if information on the basis of the generated model should be considered). Is this possible with the approach used here (running Leapfrog as an integrated part of the Python script)? Please clarify.*

**Authors' answer:**

485

The authors appreciate the referee's useful insights into the future of this work. Full integration of the probabilistic model (i.e., the "MCUP formulation") into the geologic modeling framework (i.e., Leapfrog) – or vise versa – is currently not possible. Without entering into too much detail on yet unfinished work, the authors believe that Bayesian inference can be leveraged to implement additional information in the form of subsurface observations of fault zones. The anticipated Bayesian model

490 formulation would use information from the preliminary probabilistic geomodel as the prior uncertainty, validated against new subsurface observations by way of a misfit-style likelihood function, building on the approach published by Schneeberger et al. (2017). As this future work is only in a preliminary stage of planning at the time of this writing, the authors believe its detailed treatment lies outside of the scope of the current publication.

495

The sentence has been revised to introduce a clearer stance on the possible uses of Bayesian inference following the probabilistic geomodeling laid out in this study.

**Changes to the manuscript:**

500

Future work stemming from this  input-based probabilistic geomodeling formulation may include incorporating new information in a Bayesian inference scheme to further refine the geologic model, either following the methodology introduced by de la Varga and Wellmann (2016) to infer additional information about the model parameters (requiring a full integration of the probabilistic modeling with the automated geologic modeling approach) or by Schneeberger et al. (2017) to validate the initial model in light of its generated uncertainty.

505 **3 Figures**

*overall good quality, but labels are generally far too small. Block diagrams also need orientation (at least North arrow) and axes labels should also be included.*

**Authors' answer:**

510

The necessary changes to figures throughout the text have been made as suggested by the referee.